# Robust Minimax Boosting with Performance Guarantees

**Santiago Mazuelas**[1,3]    **Verónica Álvarez**[2,1]
[1]Basque Center of Applied Mathematics (BCAM)
[2]Massachusetts Institute of Technology (MIT)
[3]IKERBASQUE-Basque Foundation for Science
smazuelas@bcamath.org, vealvar@mit.edu

## Abstract

Boosting methods often achieve excellent classification accuracy, but can experience notable performance degradation in the presence of label noise. Existing robust methods for boosting provide theoretical robustness guarantees for certain types of label noise, and can exhibit only moderate performance degradation. However, previous theoretical results do not account for realistic types of noise and finite training sizes, and existing robust methods can provide unsatisfactory accuracies, even without noise. This paper presents methods for robust minimax boosting (RMBoost) that minimize worst-case error probabilities and are robust to general types of label noise. In addition, we provide finite-sample performance guarantees for RMBoost with respect to the error obtained without noise and with respect to the best possible error (Bayes risk). The experimental results corroborate that RMBoost is not only resilient to label noise but can also provide strong classification accuracy.

## 1   Introduction

Boosting methods provide excellent predictive performance in numerous practical scenarios (see e.g., [1]). These methods determine a linear combination of base-rules through a sequential optimization process that minimizes a certain functional (often the empirical average of a convex potential). After the introduction of AdaBoost in [2], multiple boosting methods have been presented [3] together with highly-efficient implementations [4, 5]. Unfortunately, it has been widely observed that the performance of boosting methods can be significantly affected by the presence of label noise (see e.g., [6–8]). Certain boosting methods such as LogitBoost and GentleBoost provide an improved resilience to noise by using alternative convex potentials or optimization approaches [3,9]. However, as shown in [10], any boosting method that minimizes empirical averages of a convex and bounded potential can lead to poor performances in the presence of label noise (even if only a very small portion of labels are incorrect). Such a result posed a serious concern on boosting methods, as label noise is often unavoidable in practice and its extent is typically hard to quantify. For instance, in cases where an adversary intentionally modifies some labels, the machine learning practitioner may be entirely unaware of the resulting noise in the training data.

Multiple alternative boosting methods have been proposed to bypass the negative result of Long and Servedio in [10] by minimizing empirical averages of non-convex or unbounded potentials [11–15]. These robust methods can result in performances that are only mildly affected by label noise. Notably, previous theoretical results show that methods based on specific potentials (e.g., unhinged, quadratic, and sigmoid) are provably robust to certain types of label noise [13–15]. Specifically, the accuracy of these methods is not degraded by symmetric and uniform label noise for large enough training sizes. However, previous theoretical results do not show how the performance of boosting

39th Conference on Neural Information Processing Systems (NeurIPS 2025).

methods is affected by more realistic types of label noise and finite training sizes. In addition, existing robust methods do not provide theoretical guarantees with respect to the best possible error (Bayes risk). Indeed, certain robust methods have shown to achieve unsatisfactory classification performance [13, 16], since their accuracy can be low even without noise.

This paper presents robust minimax boosting (RMBoost) methods that eliminate the need to select a potential function by directly minimizing worst-case error probabilities. Our results demonstrate that RMBoost is robust to general types of label noise with finite training sizes, and can also provide strong classification performance. The main contributions presented in the paper are as follows.

- We show how RMBoost rules can be learned by solving a linear optimization problem with optimum value that corresponds to RMBoost minimax risk.
- We provide finite-sample performance guarantees for RMBoost with respect to the error obtained without noise and with respect to the Bayes risk.
- We present efficient algorithms for RMBoost learning that greedily obtain a sequence of linear combinations of base-rules with decreasing minimax risks.
- The experiments show that RMBoost can outperform existing methods in the presence of noisy labels and also achieve strong classification accuracies without noise.

*Notations:* Calligraphic letters represent sets; bold lowercase letters represent vectors; $\text{sign}(\cdot)$ denotes the sign of its argument; $\|\cdot\|_1$ and $\|\cdot\|_\infty$ denote the 1-norm and the infinity norm of its argument, respectively; $(\cdot)_+$ and $[\cdot]^\top$ denote the positive part and the transpose of its argument, respectively; $\mathbf{1}$ denotes the vector with all components equal to 1; $\preceq$ and $\succeq$ denote vector inequalities; and $\mathbb{E}_p\{\cdot\}$ and $\mathbb{V}\text{ar}_p\{\cdot\}$ denote, respectively, the expectation and variance of its argument with respect to distribution p.

## 2 Preliminaries

This section first recalls the setting for boosting methods and states the notation used in the paper. Then, we further describe related methods and results.

### 2.1 Problem formulation

Classification rules assign instances in a Borel set $\mathcal{X} \subset \mathbb{R}^d$ with labels in a finite set $\mathcal{Y}$. As is commonly done in the boosting literature, in the following we consider binary classification problems, i.e., $\mathcal{Y} = \{-1, +1\}$. We denote by $\Delta(\mathcal{X} \times \mathcal{Y})$ the set of probability distributions on $\mathcal{X} \times \mathcal{Y}$, endowed with a suitable sigma-algebra, while the set of classification rules (both deterministic and randomized) is denoted by $\text{T}(\mathcal{X}, \mathcal{Y})$. For $p \in \Delta(\mathcal{X} \times \mathcal{Y})$, we denote by $p_x \in \Delta(\mathcal{X})$ the marginal distribution over $\mathcal{X}$ and by $p(y|x)$ the conditional probability of label $y \in \mathcal{Y}$ given $x \in \mathcal{X}$. For a classification rule $h \in \text{T}(\mathcal{X}, \mathcal{Y})$, we denote by $h(y|x)$ the probability with which instance $x \in \mathcal{X}$ is assigned the label $y \in \mathcal{Y}$ (note that $h(y|x) \in \{0, 1\}$ if h is a deterministic rule). With a slight abuse of notation we denote by $h(x)$ the label assignment provided by the rule h for instance $x$, which is a random variable if h is a randomized classifier.

Supervised classification methods use training samples to obtain a classification rule h with small error probability $R(h)$, referred to as risk. If $p^* \in \Delta(\mathcal{X} \times \mathcal{Y})$ is the underlying distribution of instance-label pairs, the error probability of a classification rule $h \in \text{T}(\mathcal{X}, \mathcal{Y})$ is its expected 0-1 loss, that is, $R(h) = \mathbb{E}_{p^*}\{\ell_{0\text{-}1}(h, (x, y))\}$, where

$$\ell_{0\text{-}1}(h, (x, y)) = \mathbb{P}\{h(x) \neq y\} = 1 - h(y|x) \tag{1}$$

is the 0-1 loss of rule h at instance-label pair $(x, y)$.

The $n$ training samples $(x_1, y_1), (x_2, y_2), \ldots, (x_n, y_n)$ available for learning may be affected by label noise. We consider general types of label noise, namely, for each instance $x \in \mathcal{X}$ the label $y$ is flipped to $-y$ with a probability $0 \leq \rho_y(x) \leq 1$ for which no assumptions are imposed. Noise-less cases correspond to $\rho_{+1}(x) = \rho_{-1}(x) = 0 \; \forall \, x \in \mathcal{X}$, symmetric noise corresponds to $\rho_{+1}(x) = \rho_{-1}(x) \; \forall \, x \in \mathcal{X}$, and uniform noise corresponds to $\rho_y(x) = \rho_y(x') \; \forall \, x, x' \in \mathcal{X}, y \in \mathcal{Y}$. In practice, it is expected that the noise probabilities of most instances are rather small or zero, while those of other instances are non-negligible and unknown.

The label noise considered in the paper covers arbitrary forms of label corruption in the training samples. In particular, the results in the paper even account for deliberate manipulations of labels, where an adversary may consistently modify the labels of specific instances ($\rho_{+1}(x)$ or $\rho_{-1}(x)$ may be 1 for certain instances $x \in \mathcal{X}$ that the adversary deems most influential to learning). With noisy labels, the distribution of training samples $\mathrm{p}^{\mathrm{tr}} \in \Delta(\mathcal{X} \times \mathcal{Y})$ is different to the underlying distribution $\mathrm{p}^*$. Specifically, the marginals coincide $\mathrm{p}^{\mathrm{tr}}_x = \mathrm{p}^*_x$ while the label conditionals satisfy

$$\mathrm{p}^{\mathrm{tr}}(y|x) = (1 - \rho_y(x))\mathrm{p}^*(y|x) + \rho_{-y}(x)\mathrm{p}^*(-y|x). \tag{2}$$

Boosting methods obtain classification rules given by combinations of base-rules in a set $\mathcal{H} = \{\hbar_1, \hbar_2, \ldots, \hbar_T\} \subset \mathrm{T}(\mathcal{X}, \mathcal{Y})$. The set of base-rules considered often contains an extremely large number $T$ of simple rules, e.g., all the decision trees with a bounded number of nodes given by components of instances in the training set. Often, base-rules are themselves classification rules, i.e., $\hbar(x) \in \{-1, 1\}$ for any $x \in \mathcal{X}$. We only assume the common case in which the base-rules are bounded measurable functions $\hbar(x) \in [-1, 1]$ for any $x \in \mathcal{X}$, and that $-\hbar \in \mathcal{H}$ if $\hbar \in \mathcal{H}$.

## 2.2 Related work

Most of boosting methods can be interpreted as empirical risk minimization (ERM) techniques that learn classification rules by solving the optimization problem

$$\min_{\boldsymbol{\mu}} \frac{1}{n} \sum_{i=1}^n \phi\big(y_i \boldsymbol{\hbar}(x_i)^\top \boldsymbol{\mu}\big) \tag{3}$$

where the vector $\boldsymbol{\hbar}(x) = [\hbar_1(x), \hbar_2(x), \ldots, \hbar_T(x)]^\top$ is given by predictions of the base-rules in $\mathcal{H}$. Then, the classification rule is given by $\mathrm{h}(x) = \mathrm{sign}(\boldsymbol{\hbar}(x)^\top \boldsymbol{\mu}^*)$ with $\boldsymbol{\mu}^*$ a solution of (3). The function $\phi(\cdot)$ in (3) is referred to as potential function and its argument $y_i \boldsymbol{\hbar}(x_i)^\top \boldsymbol{\mu}$ is referred to as the margin of sample $(x_i, y_i)$ for parameters $\boldsymbol{\mu}$ (see e.g., [6]). Each potential function gives rise to a different boosting method (see e.g., [3, 6]). For instance, AdaBoost corresponds to the potential function $\phi(z) = \exp(-z)$, and LogitBoost corresponds to the potential function $\phi(z) = \log(1 + \exp(-z))$. In particular, the resilience to noise of LogitBoost is attributed to the lower values taken by the logistic potential for $z < 0$.

The results in [10] showed that even a very small fraction of noisy labels can lead to poor performances using any convex and bounded potential (i.e., $\phi(z)$ convex, $\phi'(0) < 0$, and $\lim_{z \to \infty} \phi(z) = 0$). Multiple methods have been proposed to bypass the negative result in [10] by using non-convex or unbounded potentials, such as the sigmoid potential $\phi(z) = (1 + \exp(z))^{-1}$, the quadratic potential $\phi(z) = (1 - z)^2$, and the unhinged potential $\phi(z) = 1 - z$. These potential functions have been shown to result in methods that are robust to noise in the sense that the corresponding optimization (3) is not affected by symmetric and uniform label noise for large enough training sizes [13, 14, 17]. Specifically, for some potentials including sigmoid and unhinged, the expected potential with symmetric and uniform noise is proportional to that without noise [14]. For the quadratic potential, minimizers of the expected potential with symmetric and uniform noise are equivalent to those without noise [17]. On the other hand, it has been shown that such potential functions can lead to poor classification performances, even without noise [13, 16].

The existing robustness results do not show how the performance is affected by more realistic types of noise and finite training sizes. Only the results in [14] go beyond symmetric and uniform cases and provide certain extensions of the above-described results to cases with symmetric non-uniform noise ($\rho_{+1}(x) = \rho_{-1}(x)$ varying with $x$). Furthermore, existing robustness results do not provide finite-sample generalization guarantees since they analyze the potential's actual expectation, not cases with empirical averages. In boosting methods, results for finite-sample empirical averages cannot be derived from those for actual expectations because performance bounds based on the convergence of the potential averages are inadequate for boosting (see e.g., Sec. 4.1 in [6]).

The following presents boosting methods that avoid the need to select a potential function by directly minimizing worst-case error probabilities.

## 3 Minimax boosting

RMBoost methods learn classification rules by solving the minimax problem

$$\min_{h \in T(\mathcal{X}, \mathcal{Y})} \max_{p \in \mathcal{U}} \mathbb{E}_p \big\{ \ell_{0\text{-}1}(h, (x, y)) \big\}. \tag{4}$$

Such an optimization considers general classification rules $T(\mathcal{X}, \mathcal{Y})$, probability distributions in a subset $\mathcal{U} \subset \Delta(\mathcal{X} \times \mathcal{Y})$ referred to as uncertainty set, and expected 0-1 losses (i.e., error probabilities). Minimax approaches such as that in (4) are commonly known as robust risk minimization or distributionally robust techniques [18–22]. Unlike an ERM approach, the optimization in (4) considers multiple distributions beyond the empirical distribution of training samples, so that RMBoost methods can achieve enhanced robustness as shown in the following. In addition, the optimal value of (4) referred to as the minimax risk $\overline{R}$ can be used to assess RMBoost classification error.

Unlike other distributionally robust methods, RMBoost considers uncertainty sets defined by the set of base-rules $\mathcal{H}$. Specifically, the uncertainty set of distributions $\mathcal{U}$ in (4) is given by the training samples and the base-rules $\mathcal{H}$ as

$$\mathcal{U} = \Big\{ p \in \Delta(\mathcal{X} \times \mathcal{Y}) \text{ s.t. } \big\| \mathbb{E}_p \{ y \boldsymbol{\hbar}(x) \} - \frac{1}{n} \sum_{i=1}^{n} y_i \boldsymbol{\hbar}(x_i) \big\|_{\infty} \le \lambda \Big\} \tag{5}$$

where the vector $\boldsymbol{\hbar}(x) = [\hbar_1(x), \hbar_2(x), \dots, \hbar_T(x)]^\top$ is given by the predictions of base-rules in $\mathcal{H}$ as in (3). The parameter $\lambda > 0$ accounts for the error in the finite-sample average in (5) and can be selected using standard cross-validation approaches. This selection can be enhanced taking into account the family of base-rules used or prior knowledge on the amount of label noise. In particular, more complex families of base-rules or increased levels of noise can benefit from higher values for $\lambda$. A simple default value for such parameter is $\lambda = 1/\sqrt{n}$, which is the value used in all the experimental results in the paper (Appendix H.5 further analyzes the sensitivity of the proposed methods to the choice of that hyperparameter).

The uncertainty set in (5) comprises probability distributions over instance-label pairs that are similar to the empirical distribution of training samples, as is commonly done in distributionally robust methods. While most existing methods define this similarity in terms of metrics such as the Kullback-Leibler divergence or the Wasserstein distance [18], the proposed approach defines similarity in terms of the set of base-rules considered (e.g., the set of decision trees with $t$ decision nodes). Specifically, two distributions are regarded as similar if, for any base-rule $h \in \mathcal{H}$, the expected value of $yh(x)$ changes only slightly when computed under either distribution. This notion of similarity offers two key advantages: it can yield quite restricted uncertainty sets (since common sets of base-rules are fairly expressive), and provides strong theoretical guarantees (since common sets of base-rules facilitate the fast and uniform convergence of empirical expectations).

The minimax formulation in (4) followed by RMBoost methods is particularly suitable to obtain robust classification rules since it minimizes worst-case error probabilities. However, the minimax problem in (4) may seem to be computationally prohibitive in practice. The next result shows that RMBoost classification rules can be obtained by solving the convex optimization problem

$$\min_{\boldsymbol{\mu}} F(\boldsymbol{\mu}) \coloneqq \frac{1}{2} - \frac{1}{n} \sum_{i=1}^{n} y_i \boldsymbol{\hbar}(x_i)^\top \boldsymbol{\mu} + \lambda \|\boldsymbol{\mu}\|_1 \tag{6}$$

$$\text{s.t.} \qquad -\frac{1}{2} \le \boldsymbol{\hbar}(x)^\top \boldsymbol{\mu} \le \frac{1}{2}, \ \forall x \in \mathcal{X}.$$

**Theorem 1.** If $\boldsymbol{\mu}^*$ is a solution of (6), the classification rule $h_{\boldsymbol{\mu}^*} \in T(\mathcal{X}, \mathcal{Y})$ given by

$$h_{\boldsymbol{\mu}^*}(y|x) = y \boldsymbol{\hbar}(x)^\top \boldsymbol{\mu}^* + 1/2 \tag{7}$$

is a solution of the minimax problem in (4). In addition, the minimax risk $\overline{R}$ coincides with the optimum of (6), that is $\overline{R} = F(\boldsymbol{\mu}^*)$.

*Proof.* See Appendix C. □

The result above shows that the minimax problem in (4) is equivalent to the convex optimization problem in (6). This equivalence not only provides a tractable formulation for RMBoost learning but also enables the interpretation of RMBoost methods in terms of margins. In particular, the formulation in (6) reveals that RMBoost maximizes the average margin while enforcing both upper and lower margin constraints. As a result of these constraints, an increased average margin leads to an overall increase in the distribution of margins (an average margin near $1/2$ pushes all the margins to be near $1/2$). In contrast, methods that only aim to maximize the average margin may result in instances with very low margins since others are allowed to have large margins. Hence, in existing methods the average margin is often maximized while simultaneously minimizing the margin variance [23, 24]. Other methods such as LPBoost [25] and Arc-Gv [26] that maximize the minimum margin often lead to poor classification performance since they only account for the minimum margin and not for the distribution of margins [27]. The interpretation of RMBoost in terms of margins also provides further insights for its robustness to noise. In conventional boosting methods, a sample with an incorrect label can highly impact the learning process if its margin takes a large negative value because it would result in a large potential value in (3). For methods based on quadratic or unhinged potentials, even samples with large positive margins can significantly impact the learning process because they would also result in large potential values. Such type of effects are not present in the methods proposed because the margins are bounded due to the constraints in the optimization problem (6).

Theorem 1 shows that the classification rule with the minimum worst-case error probability is given by a linear combination of base-rules. This minimax classification rule can be learned by solving the optimization (6), which carries out an L1-regularization (term $\lambda\|\boldsymbol{\mu}\|_1$) leading to a sparse combination of base-rules. The methods proposed in [28] also minimize worst-case error probabilities using a combination of base-rules. However, such work considers a transductive scenario and aims to combine a reduced set of base-rules using prior knowledge of their classification errors.

The classification rule $h_{\boldsymbol{\mu}^*}$ that minimizes the worst-case error probability randomly assigns labels with probabilities given by the predictions of base-rules, as shown in (7). Similarly to other methods (e.g., PAC-Bayes techniques [29]), it is often preferred in practice to use the corresponding deterministic classifier denoted by $h_{\boldsymbol{\mu}^*}^d$ which assigns the label corresponding to the highest probability, i.e., $h_{\boldsymbol{\mu}^*}^d(x) = \text{sign}(\boldsymbol{\hbar}(x)^\top \boldsymbol{\mu}^*)$. The error probability of the deterministic classifier is ensured to satisfy $R(h_{\boldsymbol{\mu}^*}^d) \leq 2R(h_{\boldsymbol{\mu}^*})$ (see e.g., [19, 29]) and often satisfies $R(h_{\boldsymbol{\mu}^*}^d) \leq R(h_{\boldsymbol{\mu}^*})$ in practice.

Efficient learning algorithms for RMBoost can be developed by leveraging general-purpose optimization techniques. Using as variables the positive and negative parts of $\boldsymbol{\mu}$, the optimization problem (6) is equivalent to a linear program that often has sparse solutions, as described above. Therefore, highly efficient algorithms for large-scale linear optimization can be utilized for RMBoost learning. In particular, Section 5 presents an efficient learning algorithm that address (6) using column generation methods. In addition, we next show that RMBoost does not require to solve the optimization in (6) with high accuracy, for instance the presented methods only need that the expected constraint violation in (6) is small.

As described above, the formulation of RMBoost by means of the optimization problem (6) enables to develop effective learning algorithms and also to interpret RMBoost methods in terms of margins. As shown in the following, the equivalent formulation of RMBoost in (4) as a minimax method enables to obtain performance guarantees for general types of label noise.

## 4 Generalization and robustness guarantees

This section characterizes RMBoost generalization performance with respect to the performance obtained without noise and the best possible error probability.

As shown in Theorem 1, the classification rule given by (7) minimizes the worst-case error probability if the parameter $\boldsymbol{\mu}^*$ is a solution of (6). Any other $\boldsymbol{\mu}$ can be similarly used to define classification rules as

$$h_{\boldsymbol{\mu}}(y|x) = \left[y\boldsymbol{\hbar}(x)^\top \boldsymbol{\mu} + \frac{1}{2}\right]_0^1, \;\; h_{\boldsymbol{\mu}}^d(x) = \text{sign}(\boldsymbol{\hbar}(x)^\top \boldsymbol{\mu}) \tag{8}$$

where $[\,\cdot\,]_0^1$ denotes the clip function $[\,z\,]_0^1 = (\min(z, 1))_+$.

The usage of efficient optimization algorithms for (6) can lead to suboptimal solutions that result in a value larger than the minimax risk $\overline{R}$ or fail to satisfy all the constraints. We say that $\boldsymbol{\mu}$ is an $\varepsilon_{\text{opt}}$-solution of (6) if the sum of the value suboptimality and the expected constraint violation is at most $\varepsilon_{\text{opt}}$, that is

$$\left(F(\boldsymbol{\mu}) - \overline{R}\right) + \mathbb{E}_{\text{p}_x^*}\left(|\boldsymbol{\hbar}(x)^\top \boldsymbol{\mu}| - \frac{1}{2}\right)_+ \leq \varepsilon_{\text{opt}}. \tag{9}$$

The next theorem provides generalization bounds for RMBoost with respect to the error obtained by an ideal RMBoost learned without label noise and with infinite training samples.

**Theorem 2.** Let $P_{\text{noise}}$ be the probability with which a label is incorrect at training, i.e., $P_{\text{noise}} = \mathbb{E}_{\text{p}^*}\{\rho_y(x)\}$, and $\varepsilon_{\text{est}}$ be a bound for the concentration of training averages of base-rules, that is

$$\left|\mathbb{E}_{\text{p}^{\text{tr}}}\{y\hbar(x)\} - \frac{1}{n}\sum_{i=1}^n y_i\hbar(x_i)\right| \leq \varepsilon_{\text{est}}, \ \forall\, \hbar \in \mathcal{H}. \tag{10}$$

If $\boldsymbol{\mu}$ is an $\varepsilon_{\text{opt}}$-optimal solution of (6) corresponding to $n$ training samples, and $\boldsymbol{\mu}_{\text{o}}$ is an exact solution of (6) using the exact expectation without noise $\mathbb{E}_{\text{p}^*}\{y\boldsymbol{\hbar}(x)\}$ instead of $(1/n)\sum_{i=1}^n y_i\boldsymbol{\hbar}(x_i)$. Then, we have

$$R(\text{h}_{\boldsymbol{\mu}}) \leq R(\text{h}_{\boldsymbol{\mu}_{\text{o}}}) + \varepsilon_{\text{opt}} + (\varepsilon_{\text{est}} + 2P_{\text{noise}} + \lambda)\|\boldsymbol{\mu} - \boldsymbol{\mu}_{\text{o}}\|_1. \tag{11}$$

In addition, if $P_{\text{noise}} < 1/2$, we have

$$R(\text{h}_{\boldsymbol{\mu}}) \leq R(\text{h}_{\boldsymbol{\mu}_{\text{o}}}) + \frac{\varepsilon_{\text{opt}}}{1 - 2P_{\text{noise}}} + \frac{\varepsilon_{\text{est}} + 2\sqrt{\mathbb{V}\text{ar}_{\text{p}^*}\{\rho_y(x)\}} + \lambda}{1 - 2P_{\text{noise}}}\|\boldsymbol{\mu} - \boldsymbol{\mu}_{\text{o}}\|_1. \tag{12}$$

*Proof.* See Appendix D. □

The result above shows how RMBoost error is affected by the usage of: training samples with noisy labels ($P_{\text{noise}}$), finite training sizes ($\varepsilon_{\text{est}}$), and suboptimal learning algorithms ($\varepsilon_{\text{opt}}$). The probability $P_{\text{noise}} = \mathbb{E}_{\text{p}^*}\{\rho_y(x)\}$ is rather small in common situations where most of the training labels are correct. The error term $\varepsilon_{\text{est}}$ due to the finite number of training samples can be bounded with high-probability using conventional concentration bounds (see e.g., [6, 30]). In particular, if $\mathcal{R}$ and $\mathcal{D}$ are, respectively, the Rademacher complexity and VC dimension of the family of base-rules $\mathcal{H}$, with probability at least $1 - \delta$ we have

$$\varepsilon_{\text{est}} \leq 2\mathcal{R} + \sqrt{\frac{\log 2/\delta}{2n}} \leq 2\sqrt{\frac{2\mathcal{D}\log(3n/\mathcal{D})}{n}} + \sqrt{\frac{\log 2/\delta}{2n}} \tag{13}$$

so that the sample error $\varepsilon_{\text{est}}$ generally decreases with the training size at a rate $\mathcal{O}(\sqrt{(\log n)/n})$. The error term $\varepsilon_{\text{opt}}$ remains small when appropriate algorithms for large-scale linear optimization are employed. Although problem (6) involves a large number of constraints, small expected constraint violations are sufficient to ensure a low $\varepsilon_{\text{opt}}$. In particular, the algorithm presented in the next Section achieves an $\varepsilon_{\text{opt}}$ of order $\mathcal{O}(\sqrt{(\log n)/n})$ by solving a sequence of low-dimensional linear programs.

The bound in (12) further describes how RMBoost error is affected by the non-uniformity and asymmetry of the label noise. In particular, in cases with uniform and symmetric label noise we have $\mathbb{V}\text{ar}_{\text{p}^*}\{\rho_y(x)\} = 0$, so that the bound (12) shows that RMBoost is robust to uniform and symmetric label noise. Specifically, the error of RMBoost is not affected by the presence of uniform and symmetric label noise for a large enough training size (in that case, $\varepsilon_{\text{opt}}$, $\varepsilon_{\text{est}}$, and $\lambda$ can be taken to be much smaller than $1 - 2P_{\text{noise}}$).

Differently from existing results, Theorem 2 provides performance bounds that account for finite training sizes and describe the effect of general types of label noise, including the effect due to deviations from uniform and symmetric cases (term $\mathbb{V}\text{ar}_{\text{p}^*}\{\rho_y(x)\}$). The next result provides performance guarantees for RMBoost in terms of the best possible error (Bayes risk).

**Theorem 3.** Let $h_{\text{Bayes}}$ be the Bayes rule and $\boldsymbol{\mu}_{\text{B}}$ be a parameter that satisfies

$$\sup_{x \in \mathcal{X}} \left| h_{\text{Bayes}}(x) - 2\boldsymbol{\hbar}(x)^\top \boldsymbol{\mu}_{\text{B}} \right| \leq \varepsilon_{\text{approx}}. \tag{14}$$

If $\boldsymbol{\mu}$ is an $\varepsilon_{\text{opt}}$-optimal solution of (6) corresponding to $n$ training samples possibly affected by noise, we have

$$R(h_{\boldsymbol{\mu}}) \leq R(h_{\text{Bayes}}) + \varepsilon_{\text{opt}} + \varepsilon_{\text{approx}} + (\varepsilon_{\text{est}} + 2P_{\text{noise}} + \lambda)(\|\boldsymbol{\mu} - \boldsymbol{\mu}_{\text{B}}\|_1 + \|\boldsymbol{\mu}_{\text{B}}\|_1). \tag{15}$$

*Proof.* See Appendix E. □

The result above shows that RMBoost error probability can be near the best possible performance. In particular, the bound in (15) shows that RMBoost methods are Bayes consistent in cases where combinations of base-rules can accurately approximate the Bayes rule, i.e., $\varepsilon_{\text{approx}} = 0$. Such an assumption is also required to achieve consistency with other boosting methods, as AdaBoost [31, 32], and is satisfied using common families of base-rules. For instance, any measurable function in $\mathbb{R}^d$ can be accurately approximated using trees with $d+1$ terminal nodes [31].

The results presented in this section show that RMBoost is both robust to general label noise and capable of providing near-optimal performance in common situations. The next section presents efficient algorithms for RMBoost learning.

## 5 Efficient sequential learning for RMBoost

The learning stage of RMBoost obtains parameters $\boldsymbol{\mu}$ by (approximately) solving the linear optimization problem (6). As described above, general-purpose techniques for large-scale linear optimization can be borrowed for RMBoost learning, and the following presents an efficient algorithm based on column generation methods (see e.g., [33]). These methods sequentially increase the number of variables considered and are specially effective for large-scale linear optimization since they can maintain a reduced number of variables and exploit warm-starts.

### 5.1 Learning algorithm

Algorithm 1 details the pseudocode of the presented algorithm that learns base-rules $\hbar_1, \hbar_2, \ldots, \hbar_t \in \mathcal{H}$, RMBoost parameters $\boldsymbol{\mu}^* \in \mathbb{R}^t$, and the corresponding minimax risk $R$. As in other boosting methods, the algorithm greedily selects base-rules in multiple rounds.

At each round $k \in \{1, 2, \ldots, K\}$, the algorithm uses a base learner to select a new base-rule that best fits a set of weighted samples obtained from the training samples (Step 3 in the algorithm). Then, the coefficients for the current set of selected base-rules and the weighted samples for the next round are obtained by solving the linear optimization problem (16) (Step 7 in the algorithm). In particular, the primal solution provides the coefficients for them minimax rule, the dual solution provides the next round weights, and the optimal value provides the worst-case error probability (minimax risk).

$$\min_{\boldsymbol{\mu}_+, \boldsymbol{\mu}_-} \frac{1}{2} - \frac{1}{n} \sum_{i=1}^n y_i \mathbf{u}_i^\top (\boldsymbol{\mu}_+ - \boldsymbol{\mu}_-) + \lambda \mathbf{1}^\top (\boldsymbol{\mu}_+ + \boldsymbol{\mu}_-) \quad \bigg| \quad \max_{\boldsymbol{\alpha}, \boldsymbol{\beta}} \frac{1}{2}\left(1 - \mathbf{1}^\top(\boldsymbol{\alpha} + \boldsymbol{\beta})\right)$$

$$\text{s.t.} \quad -\frac{1}{2} \leq \mathbf{u}_i^\top (\boldsymbol{\mu}_+ - \boldsymbol{\mu}_-) \leq \frac{1}{2} \quad \bigg| \quad \text{s.t.} \quad -\lambda \leq \mathbf{v}_j^\top\left(\boldsymbol{\alpha} - \boldsymbol{\beta} - \mathbf{y}/n\right) \leq \lambda$$

$$\boldsymbol{\mu}_+, \boldsymbol{\mu}_- \in \mathbb{R}^{t_k}, \boldsymbol{\mu}_+ \succeq \mathbf{0}, \boldsymbol{\mu}_- \succeq \mathbf{0} \quad \bigg| \quad \boldsymbol{\alpha}, \boldsymbol{\beta} \in \mathbb{R}^n, \boldsymbol{\alpha} \succeq \mathbf{0}, \boldsymbol{\beta} \succeq \mathbf{0}$$

$$\text{for } i = 1, 2, \ldots, n \qquad (16) \quad \bigg| \quad \text{for } j = 1, 2, \ldots, t_k \qquad (17)$$

where vectors $\mathbf{u}_i \in \mathbb{R}^{t_k}$ for $i = 1, 2, \ldots, n$ are given by $\mathbf{u}_i = [\hbar_1^{(k)}(x_i), \hbar_2^{(k)}(x_i), \ldots, \hbar_{t_k}^{(k)}(x_i)]^\top$, vectors $\mathbf{v}_j \in \mathbb{R}^n$ for $j = 1, 2, \ldots, t_k$ are given by $\mathbf{v}_j = [\hbar_j^{(k)}(x_1), \hbar_j^{(k)}(x_2), \ldots, \hbar_j^{(k)}(x_n)]^\top$, $\mathcal{H}^{(k)} = \{\hbar_1^{(k)}, \hbar_2^{(k)}, \ldots, \hbar_{t_k}^{(k)}\}$ are the $t_k$ base-rules selected at round $k$, and vector $\mathbf{y} \in \mathbb{R}^n$ is given by $\mathbf{y} = [y_1, y_2, \ldots, y_n]^\top$.

The new base-rule selected at each round (column generated in the primal) corresponds to a violated dual constraint. Specifically, each base-rule $\hbar \in \mathcal{H}$ corresponds to the dual constraints

$$-\lambda \leq [\hbar(x_1), \hbar(x_2), \ldots, \hbar(x_n)](\boldsymbol{\alpha} - \boldsymbol{\beta} - \mathbf{y}/n) \leq \lambda.$$

Hence, the most violated constraint corresponds to the base-rule that achieves

$$\max_{\hbar \in \mathcal{H}} \sum_{i=1}^{n} w_i \widetilde{y}_i \hbar(x_i) = -\min_{\hbar \in \mathcal{H}} \sum_{i=1}^{n} w_i \widetilde{y}_i \hbar(x_i) \qquad (18)$$

where the weights $\{w_i\}_{i=1}^{n}$ and labels $\{\widetilde{y}_i\}_{i=1}^{n}$ are given by

$$w_i = \left| \frac{y_i}{n} - (\alpha_i - \beta_i) \right|, \quad \widetilde{y}_i = \text{sign}\left( \frac{y_i}{n} - (\alpha_i - \beta_i) \right). \qquad (19)$$

Similarly to other boosting methods, (18) is addressed by using a base learner that returns a base-rule with small training error for samples $(x_i, \widetilde{y}_i)$ and weights $w_i$, for $i = 1, 2, \ldots, n$.

**Computational cost:** Algorithm 1 has running time and memory requirements that can be directly compared with existing boosting methods based on column generation. The complexity of Algorithm 1 is very similar to that of LPBoost [25] that also solves a linear optimization problem. Specifically, Algorithm 1 solves in each round a linear program with $2t_k$ variables and $2(t_k + n)$ constraints for $t_k$ the number of base-rules in round $k$, while LPBoost solves in each round a linear program with $n + t_k$ variables and $2n + t_k$ constraints [25]. In addition, the complexity of Algorithm 1 is lower than other methods based on column generation [34–36] that address more complicated optimization problems at each round. The complexity per round in Algorithm 1 is higher than methods such as AdaBoost or LogitBoost that do not require to solve an optimization problem in each round. However, the algorithm presented can solve such optimization problems very efficiently by leveraging the properties of column generation methods for linear problems. In particular, the previous solution can provide a valid warm-start (basic feasible solution), and previously selected base-rules can be safely removed if they correspond with strictly satisfied dual constraints [33]. The experiments in Appendix H further show that the running times of the presented method are comparable to those of existing techniques.

---

**Algorithm 1** RMBoost learning algorithm

**Input:** Training samples $\{(x_i, y_i)\}_{i=1}^{n}$, parameters $\lambda$, $K$
**Output:** $\boldsymbol{\mu}^* \in \mathbb{R}^t, \hbar_1, \hbar_2, \ldots, \hbar_t, R$

1: $\mathcal{H}^{(0)} \leftarrow \emptyset$, $R^{(0)} \leftarrow 1/2$, $\mathbf{w} \leftarrow \mathbf{1}/n$, $\widetilde{\mathbf{y}} \leftarrow \mathbf{y}$
2: **for** $k = 1, 2 \ldots, K$ **do**
3: $\quad \hbar \leftarrow \text{BaseLearner}(\mathcal{H}, \{(x_i, \widetilde{y}_i, w_i)\}_{i=1}^{n})$
4: $\quad \mathcal{H}^{(k)} \leftarrow \mathcal{H}^{(k-1)}$
5: $\quad$ **If** $\sum_{i=1}^{n} w_i \widetilde{y}_i \hbar(x_i) \leq \lambda$ BREAK **for**
6: $\quad$ Add to $\mathcal{H}^{(k)}$ the base-rule $\hbar_{t_k}^{(k)} \leftarrow \hbar$ and assign it zero coefficient
7: $\quad$ Solve (16) (warm-start $\boldsymbol{\mu}_+, \boldsymbol{\mu}_-$)
8: $\quad\quad \boldsymbol{\mu}_+, \boldsymbol{\mu}_- \leftarrow$ solution primal
9: $\quad\quad \boldsymbol{\alpha}, \boldsymbol{\beta} \leftarrow$ solution dual
10: $\quad\quad R^{(k)} \leftarrow$ optimal value
11: $\quad \boldsymbol{\mu}^{(k)} \leftarrow \boldsymbol{\mu}_+ - \boldsymbol{\mu}_-$
12: $\quad \mathbf{w} \leftarrow |\mathbf{y}/n - (\boldsymbol{\alpha} - \boldsymbol{\beta})|$
13: $\quad \widetilde{\mathbf{y}} \leftarrow \text{sign}(\mathbf{y}/n - (\boldsymbol{\alpha} - \boldsymbol{\beta}))$
14: $\quad$ **for** $j = 1, 2, \ldots, |\mathcal{H}^{(k)}|$ **do**
15: $\quad\quad$ **if** $\sum_{i=1}^{n} w_i \widetilde{y}_i \hbar_j^{(k)}(x_i) < \lambda$ **then**
16: $\quad\quad\quad$ remove $\hbar_j^{(k)}$ from $\mathcal{H}^{(k)}$
17: $R \leftarrow R^{(k)}, \boldsymbol{\mu}^* \leftarrow \boldsymbol{\mu}^{(k)}, \{\hbar_i\} \leftarrow \{\hbar_i^{(k)}\}$

---

### 5.2 Theoretical analysis

The next result provides performance guarantees for the sequence of classification rules determined by Algorithm 1.

**Theorem 4.** Let $\boldsymbol{\mu}^{(k)}$ and $R^{(k)}$ be the parameter and minimax risk determined by Algorithm 1 at round $k$. With probability at least $1 - \delta$, the error probability of the RMBoost rule at the $k$-th round satisfies

$$R(\mathrm{h}_{\boldsymbol{\mu}^{(k)}}) \leq R^{(k)} + \varepsilon(\delta) + (\varepsilon_{\text{est}} + 2P_{\text{noise}} - \lambda)\|\boldsymbol{\mu}^{(k)}\|_1 \qquad (20)$$

where $\varepsilon(\delta) = 0$ if $\|\boldsymbol{\mu}^{(k)}\|_1 \leq 1/2$, and for $\|\boldsymbol{\mu}^{(k)}\|_1 > 1/2$, $\varepsilon(\delta)$ is given by

$$\varepsilon(\delta) = 2\|\boldsymbol{\mu}^{(k)}\|_1 \sqrt{\frac{2\mathcal{D}\log(3n/\mathcal{D})}{n}} + \left(\|\boldsymbol{\mu}^{(k)}\|_1 - \frac{1}{2}\right)\sqrt{\frac{\log(1/\delta)}{2n}} \qquad (21)$$

for $\mathcal{D}$ the VC-dimension of the base-rules $\mathcal{H}$. In addition, if Algorithm 1 stops at round $k$ in Step 5 and the base learner accurately solves (18), then $\boldsymbol{\mu}^{(k)}$ is an $\varepsilon(\delta)$-optimal solution of optimization (6)

*Proof.* See Appendix F. □

Table 1: Average classification error in % ± standard deviation for RMBoost and state-of-the-art methods. The right sub-table shows cases affected by uniform and symmetric label noise with $P_{\text{noise}} = 10\%$.

| Dataset | AdaB | LogitB | XGB-Q | RMB | Minmax | AdaB | LogitB | XGB-Q | RMB | Minmax |
|---|---|---|---|---|---|---|---|---|---|---|
| Titanic | **20**±3.2 | 21±3.7 | 21±3.7 | 22±3.5 | 20±0.3 | **22**±4.1 | 23±4.5 | **22**±3.9 | **22**±3.6 | 24±0.8 |
| German | **24**±4.2 | **24**±4.5 | 25±3.4 | 27±2.8 | 26±0.6 | 30±4.2 | 29±4.4 | **27**±4.1 | **27**±5.0 | 29±0.8 |
| Blood | 24±4.4 | 27±4.3 | 22±3.9 | **20**±5.4 | 24±0.6 | 27±3.9 | 28±4.3 | 23±3.4 | **22**±3.9 | 28±0.9 |
| Credit | **14**±3.9 | **14**±4.0 | 22±5.6 | **14**±5.6 | 16±0.4 | 18±4.5 | 19±4.5 | 24±4.8 | **16**±3.8 | 21±0.8 |
| Diabet | 27±4.9 | **26**±5.3 | 34±4.6 | **26**±4.5 | 25±0.8 | 31±5.2 | 29±5.1 | 34±4.5 | **27**±5.1 | 28±1.1 |
| Raisin | 15±2.7 | 15±2.6 | 16±3.8 | **12**±3.6 | 14±0.6 | 19±3.9 | 19±4.1 | 20±3.4 | **14**±2.4 | 19±1.0 |
| QSAR | **14**±3.1 | **14**±3.1 | 23±3.5 | 15±3.1 | 17±0.5 | 19±3.5 | **18**±3.5 | 26±4.7 | 20±3.3 | 23±0.8 |
| Climat | 8.5±2.0 | 8.5±2.0 | 8.4±2.0 | **7.5**±2.0 | 9.3±0.4 | 12±2.8 | 10±2.9 | 10±3.2 | **9.5**±2.8 | 15±0.8 |

The result above shows that the error of RMBoost rules learned by Algorithm 1 is bounded by the minimax risk obtained in each round ($R^{(k)}$) together with terms that account for optimization and estimation errors ($\varepsilon(\delta)$ and $\varepsilon_{\text{est}}$) as well as the effect of noisy labels ($P_{\text{noise}}$). Due to the bound in (13), the two terms due to optimization and estimation errors decrease with the number of samples as $\mathcal{O}(\sqrt{(\log n)/n})$ and increase with the VC-dimension of the set of base rules $\mathcal{H}$. Notice that the VC-dimension of decision trees can be bounded as $\mathcal{D} \leq (2t + 1)\log_2(d + 2)$ for $t$ the number of decision nodes, and $d$ the instances' dimensionality (see e.g., [37]), leading to bounds of order $\mathcal{O}(\sqrt{(t \log d \log n)/n})$.

For other boosting methods like AdaBoost, the performance bounds that rely on VC-dimension arguments, exhibit a similar dependence on the number of samples and the VC-dimension, but increase with the number of boosting rounds (see Section 4.1 in [6]). Interestingly, the bound in (20) for RMBoost grows with $\|\boldsymbol{\mu}^{(k)}\|_1$ which can be significantly smaller than the number of rounds $k$ due to the L1-regularization imposed by $\lambda > 0$.

Theorem 4 also describes the performance guarantees of RMBoost rules determined by Algorithm 1 in terms of the results presented in Section 4. In particular, all the results in that section can be directly applied by plugging in $\varepsilon(\delta)$ as $\varepsilon_{\text{opt}}$ in cases where Algorithm 1 stops in Step 5. For general cases, Appendix G shows that the suboptimality of Algorithm 1 is increased by a term that accounts for a possible early termination and for the suboptimality of the base learner in practice.

The performance guarantees presented above reliably represent RMBoost error in practice. In particular, the experimental results below show that the minimax risk $R$ obtained by Algorithm 1 can serve to assess RMBoost prediction error.

## 6 Numerical results

The experiments compare the classification performance obtained by RMBoost with that of 8 boosting methods: the 4 state-of-the-art techniques AdaBoost [2], LogitBoost [3], GentleBoost [9], and LPBoost [25] together with the 4 robust methods RobustBoost [12], BrownBoost [11], XGBoost [4] with quadratic potential (XGB-Quad), and Robust-GBDT [38], which are specifically designed for scenarios with noisy labels. Multiple cases of label noise are evaluated using the conventional symmetric and uniform label noise ($\forall x, \rho_{+1}(x) = \rho_{-1}(x) = P_{\text{noise}}$), and also using an adversarial type of label noise ($\rho_y(x) = 1$ for the $P_{\text{noise}}$-fraction of training samples with the largest margin, $\rho_y(x) = 0$ for the other samples). This adversarial noise corresponds to label corruptions designed to maximally hinder learning by altering the most influential samples.

Due to the extensive theoretical results presented, this section remains necessarily concise. The code implementing the methods presented and reproducing the experiments can be found at `https://github.com/MachineLearningBCAM/RMBoost-NeurIPS-2025`. The supplementary materials provide additional details and results in Appendix H, including running times assessments and the results of all the boosting methods in all label noise cases.

Table 1 shows the classification error achieved by the most representative methods with 8 common datasets in noiseless cases and with symmetric and uniform label noise. The results in the table show that RMBoost can obtain state-of-the-art performance in noiseless situations and provide improved

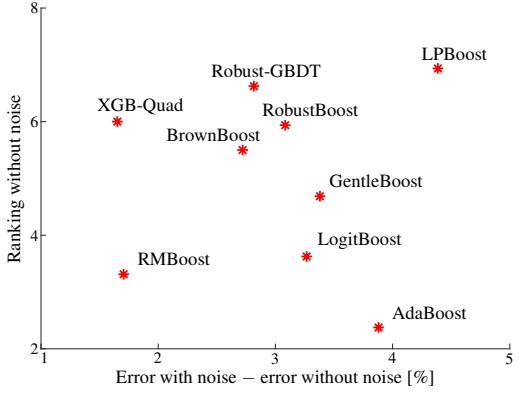

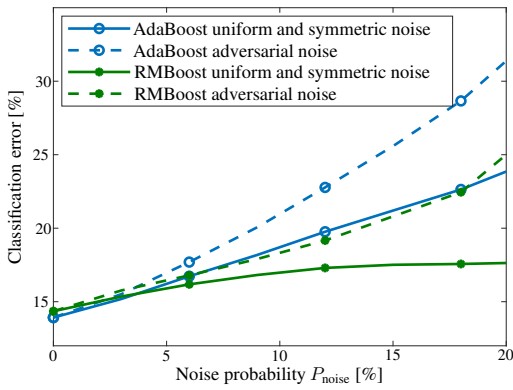

Figure 1: Trade-off classification performance vs robustness to noise in the 8 datasets (uniform and symmetric noise with $P_{\text{noise}} = 10\%$).

Figure 2: Performance degradation of AdaBoost and RMBoost methods for increased levels of noise in 'Credit' dataset.

robustness to label noise. The table also shows that the minimax risk optimized at learning is often near the RMBoost error in practice. Figure 1 summarizes the trade-off between classification performance and robustness to noise for the 9 methods in the 8 datasets. Specifically, the vertical axis describes classification performance in terms of the average ranking in the noiseless case, while the horizontal axis describes robustness to noise in terms of the average difference between the error in noisy and noiseless cases. The figure shows that RMBoost is a robust method that can also provide a strong classification performance near that of AdaBoost method.

Figure 2 further illustrates the classification performance and robustness to noise of RMBoost in comparison with AdaBoost. While Adaboost is able to achieve slightly better error with clean labels, its performance quickly deteriorates for increasing probabilities of noise, especially for adversarial noise. On the other hand, RMBoost provides strong classification accuracy on clean data that only mildly deteriorates with general types of label noise, in line with the performance guarantees presented.

## 7    Conclusion

The paper presents methods for robust minimax boosting (RMBoost) that minimize worst-case error probabilities and are robust to label noise. Differently from existing techniques, we provide finite-sample performance guarantees that describe the effect of general types of label noise as well as the Bayes consistency of RMBoost methods. In addition, the paper presents and analyzes an efficient algorithm for RMBoost learning, and experimentally shows the effectiveness of RMBoost in practice. The results in the paper show that the boosting methodology presented can enable to achieve increased levels of robustness to label noise together with strong classification performance.

**Limitations:**   The column generation approach presented in Section 5 can be directly compared with other methods such as LPBoost. However, as described above, the complexity of approaches based on column generation scales poorly with the number of training samples, compared to other methods such as AdaBoost or LogitBoost (see also experimental running times in Appendix H.3). The methodologies proposed can be implemented using alternative optimization approaches for large-scale optimization that may be more convenient computationally. The present paper focuses on the new boosting methodology proposed and the theoretical analysis of its noise robustness. Hence, we leave for future work the development of more efficient learning algorithms.

## Acknowledgements

The authors would like to thank Prof. Yoav Freund for his comments and suggestions during the development of this work. Funding in direct support of this work has been provided by project PID2022-137063NB-I00 funded by MCIN/AEI/10.13039/501100011033 and the European Union "NextGenerationEU"/PRTR, BCAM Severo Ochoa accreditation CEX2021-001142-S/MICIN/AEI/10.13039/501100011033 funded by the Ministry of Science and Innovation (Spain), and program BERC-2022-2025 funded by the Basque Government. In addition, Verónica Álvarez holds a postdoctoral grant from the Basque Government.

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

# Appendices

## A   Strong duality lemma

Some of the proofs for the results in the paper make use of Fenchel duality for linear optimization problems over probability measures. The next lemma provides the strong duality result needed for such proofs.

**Lemma 5.** Let $\mathcal{U}$ be an uncertainty set given by (5) with $\lambda > 0$. For any $\mathrm{h} \in \mathrm{T}(\mathcal{X}, \mathcal{Y})$, we have

$$\max_{\mathrm{p} \in \mathcal{U}} \mathbb{E}_{\mathrm{p}}\{\ell_{0\text{-}1}(\mathrm{h}, (x, y))\}$$

$$= \min_{\boldsymbol{\mu} \in \mathbb{R}^T} 1 - \frac{1}{n}\sum_{i=1}^{n} y_i \boldsymbol{\hbar}(x_i)^\top \boldsymbol{\mu} + \lambda\|\boldsymbol{\mu}\|_1 + \sup_{x \in \mathcal{X}, y \in \mathcal{Y}} \{y\boldsymbol{\hbar}(x)^\top \boldsymbol{\mu} - \mathrm{h}(y|x)\}. \quad (22)$$

*Proof.* In the first step of the proof, we show that the right-hand-side in (22) is the Fenchel dual of the left-hand-side. Then, the result is obtained by showing that strong duality is satisfied for the uncertainty sets used in the paper.

Let $M(\mathcal{X} \times \mathcal{Y})$ be the set of signed Borel measures over $\mathcal{X} \times \mathcal{Y}$ with bounded total variation, and $A$ be the linear mapping

$$\begin{aligned} A: \quad M(\mathcal{X} \times \mathcal{Y}) &\rightarrow \mathbb{R}^{2T+1} \\ \mathrm{p} &\mapsto \left[\int y\boldsymbol{\hbar}(x)d\mathrm{p}(x, y), -\int y\boldsymbol{\hbar}(x)d\mathrm{p}(x, y), \int d\mathrm{p}(x, y)\right]. \end{aligned} \quad (23)$$

$A$ is bounded and its adjoint operator transforms $\boldsymbol{\mu}_1, \boldsymbol{\mu}_2, \nu \in \mathbb{R}^{2T+1}$ to measurable functions over $\mathcal{X} \times \mathcal{Y}$, as $A^*(\boldsymbol{\mu}_1, \boldsymbol{\mu}_2, \nu)(x, y) = y\boldsymbol{\hbar}(x)^\top(\boldsymbol{\mu}_1 - \boldsymbol{\mu}_2) + \nu$.

Then, we have

$$\max_{\mathrm{p} \in \mathcal{U}} \mathbb{E}_{\mathrm{p}}\{\ell_{0\text{-}1}(\mathrm{h}, (x, y))\} = \max_{\mathrm{p} \in \mathcal{U}} 1 - \int \mathrm{h}(y|x)d\mathrm{p}(x, y) = 1 - \min_{\mathrm{p} \in M(\mathcal{X} \times \mathcal{Y})} f(\mathrm{p}) + g(A(\mathrm{p})) \quad (24)$$

where $f$ and $g$ are the lower semi-continuous convex functions

$$\begin{aligned} g: \quad \mathbb{R}^{2T+1} &\rightarrow \mathbb{R} \cup \{\infty\} \\ (\mathbf{a}_1, \mathbf{a}_2, b) &\mapsto \begin{cases} 0 & \text{if} \quad \mathbf{a}_1 \preceq \boldsymbol{\tau} + \lambda\mathbf{1}, \ \mathbf{a}_2 \preceq -\boldsymbol{\tau} + \lambda\mathbf{1}, b = 1 \\ \infty & \text{otherwise} \end{cases} \end{aligned} \quad (25)$$

for $\boldsymbol{\tau} = \frac{1}{n}\sum_{i=1}^{n} y_i \boldsymbol{\hbar}(x_i)$, and

$$\begin{aligned} f: \quad M(\mathcal{X} \times \mathcal{Y}) &\rightarrow \mathbb{R} \cup \{\infty\} \\ \mathrm{p} &\mapsto \begin{cases} \int \mathrm{h}(y|x)d\mathrm{p}(x, y) & \text{if} \quad \mathrm{p} \text{ is nonnegative} \\ \infty & \text{otherwise.} \end{cases} \end{aligned} \quad (26)$$

Then, the Fenchel dual (see e.g., [39]) of (24) is

$$1 - \sup_{\boldsymbol{\mu}_1, \boldsymbol{\mu}_2, \nu} -f^*(A^*(\boldsymbol{\mu}_1, \boldsymbol{\mu}_2, \nu)) - g^*(-\boldsymbol{\mu}_1, -\boldsymbol{\mu}_2, -\nu) \quad (27)$$

where $f^*$ and $g^*$ are the conjugate functions of $f$ and $g$. If $w$ is a measurable function over $\mathcal{X} \times \mathcal{Y}$, we have

$$\begin{aligned} f^*(w) &= \sup_{\mathrm{p} \succeq 0} \int (w(x, y) - \mathrm{h}(y|x))d\mathrm{p}(x, y) \\ &= \begin{cases} 0 & \text{if} \quad w(x, y) \leq \mathrm{h}(y|x), \ \forall x, y \in \mathcal{X} \times \mathcal{Y} \\ \infty & \text{otherwise} \end{cases} \end{aligned}$$

and $g^*(-\boldsymbol{\mu}_1, -\boldsymbol{\mu}_2, -\nu)$ is given by

$$\begin{aligned} \sup \quad & -\mathbf{a}_1^\top \boldsymbol{\mu}_1 - \mathbf{a}_2^\top \boldsymbol{\mu}_2 - \nu \\ \text{s.t.} \quad & \mathbf{a}_1 \preceq \boldsymbol{\tau} + \lambda\mathbf{1}, \ \mathbf{a}_2 \preceq -\boldsymbol{\tau} + \lambda\mathbf{1} \\ &= \begin{cases} -(\boldsymbol{\tau} + \lambda\mathbf{1})^\top \boldsymbol{\mu}_1 + (\boldsymbol{\tau} - \lambda\mathbf{1})^\top \boldsymbol{\mu}_2 - \nu & \text{if} \quad -\boldsymbol{\mu}_1 \succeq \mathbf{0}, -\boldsymbol{\mu}_2 \succeq \mathbf{0} \\ \infty & \text{otherwise.} \end{cases} \end{aligned}$$

Hence, the dual problem (27) becomes

$$
\begin{aligned}
&\inf_{\boldsymbol{\mu}_1,\boldsymbol{\mu}_2,\nu} \quad 1 - (\boldsymbol{\tau} + \lambda\mathbf{1})^\top \boldsymbol{\mu}_1 - (-\boldsymbol{\tau} + \lambda\mathbf{1})^\top \boldsymbol{\mu}_2 - \nu \\
&\quad\text{s.t.} \quad y\boldsymbol{\hbar}(x)^\top (\boldsymbol{\mu}_1 - \boldsymbol{\mu}_2) + \nu \leq \mathrm{h}(y|x), \ \forall (x,y) \in \mathcal{X} \times \mathcal{Y} \\
&\qquad\quad -\boldsymbol{\mu}_1 \succeq \mathbf{0}, -\boldsymbol{\mu}_2 \succeq \mathbf{0} \\
&= \inf_{\boldsymbol{\mu}_+,\boldsymbol{\mu}_-,\nu} \quad 1 + (\boldsymbol{\tau} + \lambda\mathbf{1})^\top \boldsymbol{\mu}_- + (-\boldsymbol{\tau} + \lambda\mathbf{1})^\top \boldsymbol{\mu}_+ - \nu \\
&\quad\text{s.t.} \quad y\boldsymbol{\hbar}(x)^\top (\boldsymbol{\mu}_+ - \boldsymbol{\mu}_-) + \nu \leq \mathrm{h}(y|x), \ \forall (x,y) \in \mathcal{X} \times \mathcal{Y} \\
&\qquad\quad \boldsymbol{\mu}_+ \succeq \mathbf{0}, \boldsymbol{\mu}_- \succeq \mathbf{0}
\end{aligned}
\tag{28}
$$

$$
\begin{aligned}
&= \inf_{\boldsymbol{\mu},\nu} \quad 1 - \boldsymbol{\tau}^\top \boldsymbol{\mu} + \lambda\|\boldsymbol{\mu}\|_1 - \nu \\
&\quad\text{s.t.} \quad y\boldsymbol{\hbar}(x)^\top \boldsymbol{\mu} + \nu \leq \mathrm{h}(y|x), \ \forall (x,y) \in \mathcal{X} \times \mathcal{Y}
\end{aligned}
\tag{29}
$$

$$
= \inf_{\boldsymbol{\mu}} \quad 1 - \boldsymbol{\tau}^\top \boldsymbol{\mu} + \sup_{(x,y)\in\mathcal{X}\times\mathcal{Y}} \left\{ y\boldsymbol{\hbar}(x)^\top \boldsymbol{\mu} - \mathrm{h}(y|x) \right\} + \lambda\|\boldsymbol{\mu}\|_1
\tag{30}
$$

where we have taken $\boldsymbol{\mu}_+ = -\boldsymbol{\mu}_2$, $\boldsymbol{\mu}_- = -\boldsymbol{\mu}_1$, and $\boldsymbol{\mu} = \boldsymbol{\mu}_+ - \boldsymbol{\mu}_-$. The equality in (29) is obtained from (28) because in (28) we can consider only pairs $\boldsymbol{\mu}_+, \boldsymbol{\mu}_-$ such that $\boldsymbol{\mu}_+ + \boldsymbol{\mu}_- = |\boldsymbol{\mu}_+ - \boldsymbol{\mu}_-|$ because for any pair $\boldsymbol{\mu}_+, \boldsymbol{\mu}_-$ feasible in (28), we have $\tilde{\boldsymbol{\mu}}_+ = (\boldsymbol{\mu}_+ - \boldsymbol{\mu}_-)_+$, $\tilde{\boldsymbol{\mu}}_- = (\boldsymbol{\mu}_- - \boldsymbol{\mu}_+)_+$ is a feasible pair since $\tilde{\boldsymbol{\mu}}_+ - \tilde{\boldsymbol{\mu}}_- = \boldsymbol{\mu}_+ - \boldsymbol{\mu}_-$, and we also have that $\lambda\|\tilde{\boldsymbol{\mu}}_+ - \tilde{\boldsymbol{\mu}}_-\|_1 = \lambda\mathbf{1}^\top (\tilde{\boldsymbol{\mu}}_+ + \tilde{\boldsymbol{\mu}}_-) \leq \lambda\mathbf{1}^\top (\boldsymbol{\mu}_+ + \boldsymbol{\mu}_-)$. The expression in (30) is obtained since for any feasible $(\boldsymbol{\mu}, \nu)$ in (29) we have $(\boldsymbol{\mu}, \tilde{\nu})$ is feasible if

$$
\tilde{\nu} = \inf_{(x,y)\in\mathcal{X}\times\mathcal{Y}} \left\{ \mathrm{h}(y|x) - y\boldsymbol{\hbar}(x)^\top \boldsymbol{\mu} \right\}
$$

and $\tilde{\nu} \geq \nu$.

Then, we get that (30) is at least $\max_{\mathrm{p}\in\mathcal{U}} \mathbb{E}_\mathrm{p}\{\ell_{0\text{-}1}(\mathrm{h}, (x,y))\}$ by using weak duality, next we show that strong duality (and hence equality) is achieved. Specifically, in the following we show that strong duality holds because $\mathbf{0} \in \mathrm{int}(\mathrm{dom}\, g - A\mathrm{dom}\, f)$ (see e.g., Chapter 4 in [39]), where dom denotes the set where an extended-valued function takes finite values, and int denotes the interior of a set.

We show that if $0 < \varepsilon < \lambda/(\lambda + 1 + \sqrt{T}) < 1$, the ball $B(\mathbf{0}, \varepsilon)$ with radius $\varepsilon$ centered in $\mathbf{0} \in \mathbb{R}^{2T+1}$ satisfies $B(\mathbf{0}, \varepsilon) \subset (\mathrm{dom}\, g - A\mathrm{dom}\, f) \subset \mathbb{R}^{2T+1}$. For any $\mathbf{z} \in B(\mathbf{0}, \varepsilon) \subset \mathbb{R}^{2T+1}$, there exist $\boldsymbol{\xi}_1, \boldsymbol{\xi}_2 \in B(\mathbf{0}, \lambda) \subset \mathbb{R}^T$ such that

$$
\begin{aligned}
\left( z^{(1)}, z^{(2)}, \ldots, z^{(T)} \right) &= \mathbb{E}_{\mathrm{p}_n}\{y\boldsymbol{\hbar}(x)\} + \boldsymbol{\xi}_1 - (1 - z^{(2T+1)})\mathbb{E}_{\mathrm{p}_n}\{y\boldsymbol{\hbar}(x)\} \\
\left( z^{(T+1)}, z^{(T+2)}, \ldots, z^{(2T)} \right) &= -\mathbb{E}_{\mathrm{p}_n}\{y\boldsymbol{\hbar}(x)\} + \boldsymbol{\xi}_2 + (1 - z^{(2T+1)})\mathbb{E}_{\mathrm{p}_n}\{y\boldsymbol{\hbar}(x)\} \\
z^{(2T+1)} &= 1 - (1 - z^{(2T+1)})
\end{aligned}
$$

for $\mathrm{p}_n \in \mathcal{U}$ the empirical distribution of the $n$ training samples. Such equalities are obtained because we have

$$
\begin{aligned}
\left\| \left( z^{(1)}, z^{(2)}, \ldots, z^{(T)} \right) - z^{(2T+1)}\mathbb{E}_{\mathrm{p}_n}\{y\boldsymbol{\hbar}(x)\} \right\|_2 &\leq \varepsilon(1 + \|\mathbb{E}_{\mathrm{p}_n}\{y\boldsymbol{\hbar}(x)\}\|_2) < \lambda \\
\left\| \left( z^{(T+1)}, z^{(T+2)}, \ldots, z^{(2T)} \right) + z^{(2T+1)}\mathbb{E}_{\mathrm{p}_n}\{y\boldsymbol{\hbar}(x)\} \right\|_2 &\leq \varepsilon(1 + \|\mathbb{E}_{\mathrm{p}_n}\{y\boldsymbol{\hbar}(x)\}\|_2) < \lambda
\end{aligned}
$$

since $|\hbar(x)| \leq 1$ for any $\hbar \in \mathcal{H}$.

Then, the result is obtained observing that

$$
\left( \mathbb{E}_{\mathrm{p}_n}\{y\boldsymbol{\hbar}(x)\} + \boldsymbol{\xi}_1, -\mathbb{E}_{\mathrm{p}_n}\{y\boldsymbol{\hbar}(x)\} + \boldsymbol{\xi}_2, 1 \right) \in \mathrm{dom}\, g
$$

because $\mathbb{E}_{\mathrm{p}_n} y\boldsymbol{\hbar}(x) = \boldsymbol{\tau}$. In addition, we have

$$
\left( (1 - z^{(2T+1)})\mathbb{E}_{\mathrm{p}_n}\{y\boldsymbol{\hbar}(x)\}, -(1 - z^{(2T+1)})\mathbb{E}_{\mathrm{p}_n}\{y\boldsymbol{\hbar}(x)\}, (1 - z^{(2T+1)}) \right) \in A\,\mathrm{dom}\, f
$$

because $|z^{(2T+1)}| \leq \varepsilon < 1$ and hence $(1 - z^{(2T+1)})\,\mathrm{p}_n$ is a nonnegative measure.

Finally, since strong duality holds and $\mathcal{U}$ is not empty ($\mathrm{p}_n \in \mathcal{U}$) we get that the optimal value in (22) is finite and hence the optimal in the dual is attained [39] and the 'inf' in (30) becomes 'min'.

$\square$

# B  Auxiliary Lemmas

The next lemmas provide properties that are used multiple times in the proofs below.

**Lemma 6.**

-For any classification rule $h_{\boldsymbol{\mu}}$ given by (8), we have

$$R(h_{\boldsymbol{\mu}}) \leq \frac{1}{2} - \mathbb{E}_{p^*} y \boldsymbol{\hbar}(x)^{\top} \boldsymbol{\mu} + \mathbb{E}_{p_x^*} \left( |\boldsymbol{\hbar}(x)^{\top} \boldsymbol{\mu}| - \frac{1}{2} \right)_+. \tag{31}$$

-If $p^{tr}$ is the distribution of training samples with label noise probabilities $\rho_y(x)$, for any function $f : \mathcal{X} \to \mathbb{R}$ we have

$$\mathbb{E}_{p^{tr}} \{ y f(x) \} = \mathbb{E}_{p^*} \{ y f(x)(1 - 2\rho_y(x)) \}. \tag{32}$$

*Proof.* The result in (31) is obtained because

$$h_{\boldsymbol{\mu}}(y|x) = \left[ y\boldsymbol{\hbar}(x)^{\top} \boldsymbol{\mu} + \frac{1}{2} \right]_0^1 = y\boldsymbol{\hbar}(x)^{\top} \boldsymbol{\mu} + \frac{1}{2} - y \left( \boldsymbol{\hbar}(x)^{\top} \boldsymbol{\mu} - \frac{1}{2} \right)_+ + y \left( -\boldsymbol{\hbar}(x)^{\top} \boldsymbol{\mu} - \frac{1}{2} \right)_+$$

as a consequence of the definition of $h_{\boldsymbol{\mu}}$ in (8). Therefore, we have

$$h_{\boldsymbol{\mu}}(y|x) \geq y\boldsymbol{\hbar}(x)^{\top} \boldsymbol{\mu} + \frac{1}{2} - \left( |\boldsymbol{\hbar}(x)^{\top} \boldsymbol{\mu}| - \frac{1}{2} \right)_+, \ \forall x \in \mathcal{X}, y \in \mathcal{Y} \tag{33}$$

that directly leads to (31) since $R(h_{\boldsymbol{\mu}}) = \mathbb{E}_{p^*} \{ 1 - h(y|x) \}$.

The result in (32) is directly obtained because using (2) we get

$$\begin{aligned}
\mathbb{E}_{p^{tr}} \{ y f(x) \} &= \mathbb{E}_{p_x^*} \{ f(x) \big( (1 - \rho_{+1}(x)) p^*(+1|x) + \rho_{-1}(x) p^*(-1|x) \big) \} \\
&\quad + \mathbb{E}_{p_x^*} \{ -f(x) \big( (1 - \rho_{-1}(x)) p^*(-1|x) + \rho_{+1}(x) p^*(+1|x) \big) \} \\
&= \mathbb{E}_{p^*} \{ y f(x)(1 - 2\rho_y(x)) \}.
\end{aligned}$$

$\square$

**Lemma 7.** Let $\varepsilon_{est}$ be given as in (10) and $P_{noise}$ be the probability with which a label is incorrect at training. If $\boldsymbol{\mu}$ is an $\varepsilon_{opt}$-solution of (6), we have

$$R(h_{\boldsymbol{\mu}}) \leq \overline{R} + \varepsilon_{opt} + (\varepsilon_{est} + 2P_{noise} - \lambda) \| \boldsymbol{\mu} \|_1. \tag{34}$$

*Proof.* Using (31) in Lemma 6 above, we get

$$\begin{aligned}
R(h_{\boldsymbol{\mu}}) &\leq F(\boldsymbol{\mu}) + \mathbb{E}_{p_x^*} \left( |\boldsymbol{\hbar}(x)^{\top} \boldsymbol{\mu}| - \frac{1}{2} \right)_+ - \lambda \| \boldsymbol{\mu} \|_1 + \left( \boldsymbol{\tau} - \mathbb{E}_{p^*} y \boldsymbol{\hbar}(x) \right)^{\top} \boldsymbol{\mu} \\
&\leq \overline{R} + \varepsilon_{opt} + |\mathbb{E}_{p^*} y \boldsymbol{\hbar}(x) - \boldsymbol{\tau}|^{\top} |\boldsymbol{\mu}| - \lambda \| \boldsymbol{\mu} \|_1
\end{aligned}$$

after adding and subtracting $\lambda \| \boldsymbol{\mu} \|_1$ and $\boldsymbol{\tau}^{\top} \boldsymbol{\mu}$ with $\boldsymbol{\tau} = \frac{1}{n} \sum_{i=1}^n y_i \boldsymbol{\hbar}(x_i)$. Therefore, the result is obtained by using Hölder's inequality because we have

$$\| \mathbb{E}_{p^*} y \boldsymbol{\hbar}(x) - \boldsymbol{\tau} \|_{\infty} \leq \left\| \mathbb{E}_{p^{tr}} y \boldsymbol{\hbar}(x) - \frac{1}{n} \sum_{i=1}^n y_i \boldsymbol{\hbar}(x_i) \right\|_{\infty} + \left\| \mathbb{E}_{p^*} y \boldsymbol{\hbar}(x) - \mathbb{E}_{p^{tr}} y \boldsymbol{\hbar}(x) \right\|_{\infty} \tag{35}$$

$$\leq \varepsilon_{est} + 2\mathbb{E}_{p^*} \{ \rho_y(x) \} = \varepsilon_{est} + 2P_{noise} \tag{36}$$

by using (32) in Lemma 6 and the fact that $|\hbar(x)| \leq 1 \ \forall \hbar \in \mathcal{H}$. $\square$

# C  Proof of Theorem 1

Using Lemma 5 in Appendix A and taking $\boldsymbol{\tau} = \frac{1}{n}\sum_{i=1}^{n} y_i \boldsymbol{\hbar}(x_i)$, we have

$$\min_{h \in T(\mathcal{X},\mathcal{Y})} \max_{p \in \mathcal{U}} \mathbb{E}_p\{\ell_{0\text{-}1}(h,(x,y))\} = \min_{h,\boldsymbol{\mu}} 1 - \boldsymbol{\tau}^\top \boldsymbol{\mu} + \lambda \|\boldsymbol{\mu}\|_1 + \sup_{x \in \mathcal{X}, y \in \mathcal{Y}} \{y\boldsymbol{\hbar}(x)^\top \boldsymbol{\mu} - h(y|x)\}$$

$$= \min_{\boldsymbol{\mu}} 1 - \boldsymbol{\tau}^\top \boldsymbol{\mu} + \lambda \|\boldsymbol{\mu}\|_1 + \min_{h} \sup_{x \in \mathcal{X}, y \in \mathcal{Y}} \{y\boldsymbol{\hbar}(x)^\top \boldsymbol{\mu} - h(y|x)\}$$

$$(37)$$

and

$$\min_{h} \sup_{x \in \mathcal{X}, y \in \mathcal{Y}} \{y\boldsymbol{\hbar}(x)^\top \boldsymbol{\mu} - h(y|x)\} = \min_{h,\nu} \; \nu$$
$$\text{s.t.} \quad y\boldsymbol{\hbar}(x)^\top \boldsymbol{\mu} - h(y|x) \le \nu, \; \forall x \in \mathcal{X}, y \in \mathcal{Y}.$$

In addition, we have

$$y\boldsymbol{\hbar}(x)^\top \boldsymbol{\mu} - h(y|x) \le \nu, \forall x \in \mathcal{X}, y \in \mathcal{Y} \Rightarrow \nu \ge \sup_{x \in \mathcal{X}} \max\left\{\boldsymbol{\hbar}(x)^\top \boldsymbol{\mu} - 1, -\boldsymbol{\hbar}(x)^\top \boldsymbol{\mu} - 1, -\frac{1}{2}\right\}$$

since $h(y|x) \le 1$ and $h(1|x) + h(-1|x) = 1$ for any $x \in \mathcal{X}, y \in \mathcal{Y}$. For each $\boldsymbol{\mu}$, we first prove that there exists a classification rule $h$ satisfying

$$h(y|x) \ge y\boldsymbol{\hbar}(x)^\top \boldsymbol{\mu} - \sup_{x \in \mathcal{X}} \max\left\{\boldsymbol{\hbar}(x)^\top \boldsymbol{\mu} - 1, -\boldsymbol{\hbar}(x)^\top \boldsymbol{\mu} - 1, -\frac{1}{2}\right\}. \qquad (38)$$

Clearly, we have

$$\sup_{x \in \mathcal{X}} \max\left\{\boldsymbol{\hbar}(x)^\top \boldsymbol{\mu} - 1, -\boldsymbol{\hbar}(x)^\top \boldsymbol{\mu} - 1, -\frac{1}{2}\right\} \ge -\frac{1}{2}$$

$$\sup_{x \in \mathcal{X}} \max\left\{\boldsymbol{\hbar}(x)^\top \boldsymbol{\mu} - 1, -\boldsymbol{\hbar}(x)^\top \boldsymbol{\mu} - 1, -\frac{1}{2}\right\} \ge y\boldsymbol{\hbar}(x)^\top \boldsymbol{\mu} - 1, \; \forall x \in \mathcal{X}, y \in \mathcal{Y}.$$

Therefore, there exists a classification rule satisfying (38) because

$$\sum_{y \in \mathcal{Y}} \left(y\boldsymbol{\hbar}(x)^\top \boldsymbol{\mu} - \sup_{x \in \mathcal{X}} \max\left\{\boldsymbol{\hbar}(x)^\top \boldsymbol{\mu} - 1, -\boldsymbol{\hbar}(x)^\top \boldsymbol{\mu} - 1, -\frac{1}{2}\right\}\right) \le \sum_{y \in \mathcal{Y}} y\boldsymbol{\hbar}(x)^\top \boldsymbol{\mu} + \frac{1}{2} = 1, \forall x \in \mathcal{X}$$

and

$$y\boldsymbol{\hbar}(x)^\top \boldsymbol{\mu} - \sup_{x \in \mathcal{X}} \max\left\{\boldsymbol{\hbar}(x)^\top \boldsymbol{\mu} - 1, -\boldsymbol{\hbar}(x)^\top \boldsymbol{\mu} - 1, -\frac{1}{2}\right\} \le 1, \forall x \in \mathcal{X}, y \in \mathcal{Y}.$$

Then, such rules are solutions of

$$\min_{h} \sup_{x \in \mathcal{X}, y \in \mathcal{Y}} \{y\boldsymbol{\hbar}(x)^\top \boldsymbol{\mu} - h(y|x)\} = \sup_{x \in \mathcal{X}} \max\left\{\boldsymbol{\hbar}(x)^\top \boldsymbol{\mu} - 1, -\boldsymbol{\hbar}(x)^\top \boldsymbol{\mu} - 1, -\frac{1}{2}\right\}$$

because for any $h \in T(\mathcal{X}, \mathcal{Y})$ we have

$$\sup_{x \in \mathcal{X}, y \in \mathcal{Y}} \{y\boldsymbol{\hbar}(x)^\top \boldsymbol{\mu} - h(y|x)\} = \sup_{x \in \mathcal{X}} \max\{\boldsymbol{\hbar}(x)^\top \boldsymbol{\mu} - h(1|x), -\boldsymbol{\hbar}(x)^\top \boldsymbol{\mu} - h(-1|x)\}$$

$$\ge \sup_{x \in \mathcal{X}} \max\left\{\boldsymbol{\hbar}(x)^\top \boldsymbol{\mu} - h(1|x), -\boldsymbol{\hbar}(x)^\top \boldsymbol{\mu} - h(-1|x), -\frac{1}{2}\right\}$$

$$\ge \sup_{x \in \mathcal{X}} \max\left\{\boldsymbol{\hbar}(x)^\top \boldsymbol{\mu} - 1, -\boldsymbol{\hbar}(x)^\top \boldsymbol{\mu} - 1, -\frac{1}{2}\right\}$$

since

$$-\frac{1}{2} = \frac{1}{2}\left(\boldsymbol{\hbar}(x)^\top \boldsymbol{\mu} - h(1|x)\right) + \frac{1}{2}\left(-\boldsymbol{\hbar}(x)^\top \boldsymbol{\mu} - h(-1|x)\right)$$
$$\le \max\{\boldsymbol{\hbar}(x)^\top \boldsymbol{\mu} - h(1|x), -\boldsymbol{\hbar}(x)^\top \boldsymbol{\mu} - h(-1|x)\}$$

and if $h$ satisfies (38), we get

$$\sup_{x \in \mathcal{X}, y \in \mathcal{Y}} \{y\boldsymbol{\hbar}(x)^\top \boldsymbol{\mu} - h(y|x)\} \le \sup_{x \in \mathcal{X}} \max\left\{\boldsymbol{\hbar}(x)^\top \boldsymbol{\mu} - 1, -\boldsymbol{\hbar}(x)^\top \boldsymbol{\mu} - 1, -\frac{1}{2}\right\}.$$

Therefore, we have that (37) becomes

$$\min_{h \in T(\mathcal{X}, \mathcal{Y})} \max_{p \in \mathcal{U}} \mathbb{E}_p\{\ell_{0\text{-}1}(h, (x, y))\} = \min_{\boldsymbol{\mu}} 1 - \boldsymbol{\tau}^\top \boldsymbol{\mu} + \lambda \|\boldsymbol{\mu}\|_1 \qquad (39)$$

$$+ \sup_{x \in \mathcal{X}} \max \left\{ \boldsymbol{\hbar}(x)^\top \boldsymbol{\mu} - 1, -\boldsymbol{\hbar}(x)^\top \boldsymbol{\mu} - 1, -\frac{1}{2} \right\}$$

and the result is obtained by observing that if $\boldsymbol{\mu}^*$ is a solution of (39), it has to satisfy

$$-\frac{1}{2} \le \boldsymbol{\hbar}(x)^\top \boldsymbol{\mu}^* \le \frac{1}{2}, \ \forall x \in \mathcal{X}.$$

Otherwise, we would have that

$$\sup_{x \in \mathcal{X}} \max \left\{ \boldsymbol{\hbar}(x)^\top \boldsymbol{\mu}^* - 1, -\boldsymbol{\hbar}(x)^\top \boldsymbol{\mu}^* - 1, -\frac{1}{2} \right\} = \sup_{x \in \mathcal{X}} \max \left\{ \boldsymbol{\hbar}(x)^\top \boldsymbol{\mu}^* - 1, -\boldsymbol{\hbar}(x)^\top \boldsymbol{\mu}^* - 1 \right\} = \frac{C}{2} - 1$$

with $C > 1$. Then taking $\widetilde{\boldsymbol{\mu}} = \boldsymbol{\mu}^*/C$ the objective of (39) at such $\widetilde{\boldsymbol{\mu}}$ would become

$$-\boldsymbol{\tau}^\top \frac{\boldsymbol{\mu}^*}{C} + \lambda \left\| \frac{\boldsymbol{\mu}^*}{C} \right\|_1 + \max \left\{ \sup_{x \in \mathcal{X}} \max\{\boldsymbol{\hbar}(x)^\top \frac{\boldsymbol{\mu}^*}{C}, -\boldsymbol{\hbar}(x)^\top \frac{\boldsymbol{\mu}^*}{C}\}, \frac{1}{2} \right\}$$

$$= -\boldsymbol{\tau}^\top \frac{\boldsymbol{\mu}^*}{C} + \lambda \left\| \frac{\boldsymbol{\mu}^*}{C} \right\|_1 + \max \left\{ \frac{1}{C} \sup_{x \in \mathcal{X}} \max\{\boldsymbol{\hbar}(x)^\top \boldsymbol{\mu}^*, -\boldsymbol{\hbar}(x)^\top \boldsymbol{\mu}^*\}, \frac{1}{2} \right\}$$

$$= \frac{1}{C} \left( -\boldsymbol{\tau}^\top \boldsymbol{\mu}^* + \lambda \|\boldsymbol{\mu}^*\|_1 + \frac{C}{2} \right)$$

and hence the value at $\widetilde{\boldsymbol{\mu}}$ would be smaller than that at $\boldsymbol{\mu}^*$ since $C > 1$ and the optimum value in (39) is positive, which is in contradiction with $\boldsymbol{\mu}^*$ being a solution of (39). Then, $h_{\boldsymbol{\mu}^*}$ in (7) and $\overline{R} = F(\boldsymbol{\mu}^*)$ are, respectively, a solution and the optimum value of (4) as a direct consequence of (37).

## D  Proof of Theorem 2

Using (31) in Lemma 6 above, we have

$$R(h_{\boldsymbol{\mu}}) \le \frac{1}{2} - \mathbb{E}_{p^*}\{y\boldsymbol{\hbar}(x)^\top \boldsymbol{\mu}\} - F(\boldsymbol{\mu}) + \overline{R} + \varepsilon_{\text{opt}}$$

so that, using the definition of the minimax risk $\overline{R}$ and the function $F(\boldsymbol{\mu})$ in (6), we get

$$R(h_{\boldsymbol{\mu}}) \le \varepsilon_{\text{opt}} + \frac{1}{n} \sum_{i=1}^n y_i \boldsymbol{\hbar}(x_i)^\top \boldsymbol{\mu} - \mathbb{E}_{p^*}\{y\boldsymbol{\hbar}(x)^\top \boldsymbol{\mu}\} - \lambda\|\boldsymbol{\mu}\|_1 + \frac{1}{2} - \frac{1}{n} \sum_{i=1}^n y_i \boldsymbol{\hbar}(x_i)^\top \boldsymbol{\mu}_\text{o} + \lambda\|\boldsymbol{\mu}_\text{o}\|_1$$

hence, adding and subtracting $\mathbb{E}_{p^*}\{y\boldsymbol{\hbar}(x)^\top \boldsymbol{\mu}_\text{o}\}$, we get

$$R(h_{\boldsymbol{\mu}}) \le R(h_{\boldsymbol{\mu}_\text{o}}) + \varepsilon_{\text{opt}} + \left( \frac{1}{n} \sum_{i=1}^n y_i \boldsymbol{\hbar}(x_i) - \mathbb{E}_{p^*}\{y\boldsymbol{\hbar}(x)\} \right)^\top (\boldsymbol{\mu} - \boldsymbol{\mu}_\text{o}) - \lambda\|\boldsymbol{\mu}\|_1 + \lambda\|\boldsymbol{\mu}_\text{o}\|_1$$

since $R(h_{\boldsymbol{\mu}_\text{o}}) = 1/2 - \mathbb{E}_{p^*}\{y\boldsymbol{\hbar}(x)^\top \boldsymbol{\mu}_\text{o}\}$. Therefore, the result in (11) is obtained using the reverse triangular inequality together with Hölder's inequality and the fact that

$$\left\| \mathbb{E}_{p^*}\{y\boldsymbol{\hbar}(x)\} - \frac{1}{n} \sum_{i=1}^n y_i \boldsymbol{\hbar}(x_i) \right\|_\infty$$

$$\le \left\| \mathbb{E}_{p^{\text{tr}}}\{y\boldsymbol{\hbar}(x)\} - \frac{1}{n} \sum_{i=1}^n y_i \boldsymbol{\hbar}(x_i) \right\|_\infty + \|\mathbb{E}_{p^*}\{y\boldsymbol{\hbar}(x)\} - \mathbb{E}_{p^{\text{tr}}}\{y\boldsymbol{\hbar}(x)\}\|_\infty$$

$$\le \varepsilon_{\text{est}} + 2P_{\text{noise}}$$

using (32) in Lemma 6.

For the second result, we have

$$R(h_{\boldsymbol{\mu}}) \le \frac{1}{2} - \mathbb{E}_{p^*}\{y\boldsymbol{\hbar}(x)^\top \boldsymbol{\mu}\} + \mathbb{E}_{p^*}\left( |\boldsymbol{\hbar}(x)^\top \boldsymbol{\mu}| - \frac{1}{2} \right)_+$$

using (31) in Lemma 6. Then, adding and subtracting $(\overline{R} - 1/2)/(1 - 2P_{\text{noise}})$ we get

$$
R(\mathrm{h}_{\boldsymbol{\mu}}) \leq -\,\mathbb{E}_{\mathrm{p}^*}\{y\boldsymbol{\hbar}(x)^\top\boldsymbol{\mu}\} + \mathbb{E}_{\mathrm{p}^*}\left(\left|\boldsymbol{\hbar}(x)^\top\boldsymbol{\mu}\right| - \frac{1}{2}\right)_+ - \frac{1}{n}\sum_{i=1}^n \frac{y_i\boldsymbol{\hbar}(x_i)^\top\boldsymbol{\mu}_{\mathrm{o}}}{1 - 2P_{\text{noise}}} + \frac{\lambda}{1 - 2P_{\text{noise}}}\|\boldsymbol{\mu}_{\mathrm{o}}\|_1
$$

$$
+ \frac{1}{2} + \frac{1}{n}\sum_{i=1}^n \frac{y_i\boldsymbol{\hbar}(x_i)^\top\boldsymbol{\mu}}{1 - 2P_{\text{noise}}} - \frac{\lambda}{1 - 2P_{\text{noise}}}\|\boldsymbol{\mu}\|_1 + \frac{\varepsilon_{\text{opt}}}{1 - 2P_{\text{noise}}} - \frac{\mathbb{E}_{\mathrm{p}^*}\left(\left|\boldsymbol{\hbar}(x)^\top\boldsymbol{\mu}\right| - \frac{1}{2}\right)_+}{1 - 2P_{\text{noise}}}
\tag{40}
$$

using the definition of $\boldsymbol{\mu}$ and $\boldsymbol{\mu}_{\mathrm{o}}$ and the fact that

$$
-\frac{\overline{R} - 1/2}{1 - 2P_{\text{noise}}} \leq \frac{\varepsilon_{\text{opt}}}{1 - 2P_{\text{noise}}} - \frac{\mathbb{E}_{\mathrm{p}^*}\left(\left|\boldsymbol{\hbar}(x)^\top\boldsymbol{\mu}\right| - \frac{1}{2}\right)_+}{1 - 2P_{\text{noise}}} + \frac{-F(\boldsymbol{\mu}) + 1/2}{1 - 2P_{\text{noise}}}.
$$

Grouping terms in (40) and using the fact that $R(\mathrm{h}_{\boldsymbol{\mu}_{\mathrm{o}}}) = 1/2 - \mathbb{E}_{\mathrm{p}^*}\{y\boldsymbol{\hbar}(x)^\top\boldsymbol{\mu}_{\mathrm{o}}\}$, we get

$$
R(\mathrm{h}_{\boldsymbol{\mu}}) \leq R(\mathrm{h}_{\boldsymbol{\mu}_{\mathrm{o}}}) + \frac{\varepsilon_{\text{opt}}}{1 - 2P_{\text{noise}}} + \left(\frac{1}{n}\sum_{i=1}^n \frac{y_i\boldsymbol{\hbar}(x_i)}{1 - 2P_{\text{noise}}} - \mathbb{E}_{\mathrm{p}^*}\{y\boldsymbol{\hbar}(x)\}\right)^\top(\boldsymbol{\mu} - \boldsymbol{\mu}_{\mathrm{o}})
$$

$$
+ \frac{\lambda}{1 - 2P_{\text{noise}}}(\|\boldsymbol{\mu}_{\mathrm{o}}\|_1 - \|\boldsymbol{\mu}\|_1)
$$

so that the result is obtained using the reverse triangle inequality together with Hölder's inequality and the bound

$$
\left\|\frac{\mathbb{E}_{\mathrm{p}^{\text{tr}}}\{y\boldsymbol{\hbar}(x)\}}{1 - 2P_{\text{noise}}} - \mathbb{E}_{\mathrm{p}^*}\{y\boldsymbol{\hbar}(x)\}\right\|_\infty = \left\|2\frac{\mathbb{E}_{\mathrm{p}^*}\{y\boldsymbol{\hbar}(x)(\rho_y(x) - P_{\text{noise}})\}}{1 - 2P_{\text{noise}}}\right\|_\infty \leq 2\frac{\sqrt{\mathbb{V}\mathrm{ar}_{\mathrm{p}^*}\{\rho_y(x)\}}}{1 - 2P_{\text{noise}}}
$$

that follows using (32), $P_{\text{noise}} = \mathbb{E}_{\mathrm{p}^*}\{\rho_y(x)\}$, Jensen inequality, and the fact that $|\hbar(x)| \leq 1\ \forall \hbar \in \mathcal{H}$.

# E   Proof of Theorem 3

Using (31) in Lemma 6, we have

$$
R(\mathrm{h}_{\boldsymbol{\mu}}) \leq \frac{1}{2} - \mathbb{E}_{\mathrm{p}^*}\{y\boldsymbol{\hbar}(x)\}^\top\boldsymbol{\mu} - F(\boldsymbol{\mu}) + \overline{R} + \varepsilon_{\text{opt}}.
$$

If $C = \max(1, \sup_{x \in \mathcal{X}} |2\boldsymbol{\hbar}(x)^\top\boldsymbol{\mu}_{\mathrm{B}}|)$, the vector $\boldsymbol{\mu}_{\mathrm{B}}/C$ is feasible for (6), so that using the definition of the minimax risk $\overline{R}$ and the function $F(\boldsymbol{\mu})$ in (6), we get

$$
R(\mathrm{h}_{\boldsymbol{\mu}}) \leq \varepsilon_{\text{opt}} + \frac{1}{n}\sum_{i=1}^n y_i\boldsymbol{\hbar}(x_i)^\top\boldsymbol{\mu} - \mathbb{E}_{\mathrm{p}^*}\{y\boldsymbol{\hbar}(x)\}^\top\boldsymbol{\mu} - \lambda\|\boldsymbol{\mu}\|_1 + \frac{1}{2} - \frac{1}{n}\sum_{i=1}^n y_i\boldsymbol{\hbar}(x_i)^\top\frac{\boldsymbol{\mu}_{\mathrm{B}}}{C} + \frac{\lambda}{C}\|\boldsymbol{\mu}_{\mathrm{B}}\|_1.
$$

Hence, adding and subtracting $\mathbb{E}_{\mathrm{p}^*}\{y\mathrm{h}_{\text{Bayes}}(x)\}/2$ and $\mathbb{E}_{\mathrm{p}^*}\{y\boldsymbol{\hbar}(x)\}^\top\boldsymbol{\mu}_{\mathrm{B}}/C$, we get

$$
R(\mathrm{h}_{\boldsymbol{\mu}}) \leq R(\mathrm{h}_{\text{Bayes}}) + \varepsilon_{\text{opt}} + \left(\frac{1}{n}\sum_{i=1}^n y_i\boldsymbol{\hbar}(x_i) - \mathbb{E}_{\mathrm{p}^*}\{y\boldsymbol{\hbar}(x)\}\right)^\top\left(\boldsymbol{\mu} - \frac{\boldsymbol{\mu}_{\mathrm{B}}}{C}\right) - \lambda\|\boldsymbol{\mu}\|_1
$$

$$
+ \frac{\lambda}{C}\|\boldsymbol{\mu}_{\mathrm{B}}\|_1 + \mathbb{E}_{\mathrm{p}^*}\left\{\frac{y\mathrm{h}_{\text{Bayes}}(x)}{2} - \frac{y\boldsymbol{\hbar}(x)^\top\boldsymbol{\mu}_{\mathrm{B}}}{C}\right\}
\tag{41}
$$

since $R(\mathrm{h}_{\mathrm{Bayes}}) = 1/2 - \mathbb{E}_{\mathrm{p}^*}\{y\mathrm{h}_{\mathrm{Bayes}}(x)\}/2$. For the last term in (41), we have

$$\left|\mathbb{E}_{\mathrm{p}^*}\left\{\frac{y\mathrm{h}_{\mathrm{Bayes}}(x)}{2} - y\boldsymbol{\hbar}(x)^\top\frac{\boldsymbol{\mu}_{\mathrm{B}}}{C}\right\}\right| = \left|\int\left(\frac{\mathrm{h}_{\mathrm{Bayes}}(x)}{2} - \boldsymbol{\hbar}(x)^\top\frac{\boldsymbol{\mu}_{\mathrm{B}}}{C}\right)y d\mathrm{p}^*(x,y)\right|$$

$$\leq \frac{1}{2}\sup_{x\in\mathcal{X}}\left|\mathrm{h}_{\mathrm{Bayes}}(x) - \frac{2\boldsymbol{\hbar}(x)^\top\boldsymbol{\mu}_{\mathrm{B}}}{C}\right|$$

$$\leq \frac{1}{2}\sup_{x\in\mathcal{X}}|\mathrm{h}_{\mathrm{Bayes}}(x) - 2\boldsymbol{\hbar}(x)^\top\boldsymbol{\mu}_{\mathrm{B}}|$$

$$+ \frac{1}{2}\sup_{x\in\mathcal{X}}\left|\frac{2\boldsymbol{\hbar}(x)^\top\boldsymbol{\mu}_{\mathrm{B}}}{C} - 2\boldsymbol{\hbar}(x)^\top\boldsymbol{\mu}_{\mathrm{B}}\right|$$

$$\leq \frac{\varepsilon_{\mathrm{approx}}}{2} + \frac{1}{2}\sup_{x\in\mathcal{X}}|2\boldsymbol{\hbar}(x)^\top\boldsymbol{\mu}_{\mathrm{B}}|\left(1 - \frac{1}{C}\right)$$

$$= \frac{\varepsilon_{\mathrm{approx}}}{2} + \frac{1}{2}(C-1) \leq \varepsilon_{\mathrm{approx}} \tag{42}$$

where (42) is obtained because $C = 1$ or

$$C = \sup_{x\in\mathcal{X}}|2\boldsymbol{\hbar}(x)^\top\boldsymbol{\mu}_{\mathrm{B}}| \leq \sup_{x\in\mathcal{X}}|2\boldsymbol{\hbar}(x)^\top\boldsymbol{\mu}_{\mathrm{B}} - \mathrm{h}_{\mathrm{Bayes}}(x)| + \sup_{x\in\mathcal{X}}|\mathrm{h}_{\mathrm{Bayes}}(x)| \leq \varepsilon_{\mathrm{approx}} + 1.$$

For the third term in (41), using Hölder's inequality we get

$$\left(\frac{1}{n}\sum_{i=1}^n y_i\boldsymbol{\hbar}(x_i) - \mathbb{E}_{\mathrm{p}^*}\{y\boldsymbol{\hbar}(x)\}\right)^\top\left(\boldsymbol{\mu} - \frac{\boldsymbol{\mu}_{\mathrm{B}}}{C}\right) \leq \left\|\mathbb{E}_{\mathrm{p}^*}\{y\boldsymbol{\hbar}(x)\} - \frac{1}{n}\sum_{i=1}^n y_i\boldsymbol{\hbar}(x_i)\right\|_\infty\left\|\boldsymbol{\mu} - \frac{\boldsymbol{\mu}_{\mathrm{B}}}{C}\right\|_1$$

$$\leq \left\|\mathbb{E}_{\mathrm{p}^{\mathrm{tr}}}\{y\boldsymbol{\hbar}(x)\} - \frac{1}{n}\sum_{i=1}^n y_i\boldsymbol{\hbar}(x_i)\right\|_\infty\left\|\boldsymbol{\mu} - \frac{\boldsymbol{\mu}_{\mathrm{B}}}{C}\right\|_1$$

$$+ \left\|\mathbb{E}_{\mathrm{p}^*}\{y\boldsymbol{\hbar}(x)\} - \mathbb{E}_{\mathrm{p}^{\mathrm{tr}}}\{y\boldsymbol{\hbar}(x)\}\right\|_\infty\left\|\boldsymbol{\mu} - \frac{\boldsymbol{\mu}_{\mathrm{B}}}{C}\right\|_1$$

$$\leq (\varepsilon_{\mathrm{est}} + 2P_{\mathrm{noise}})\left\|\boldsymbol{\mu} - \frac{\boldsymbol{\mu}_{\mathrm{B}}}{C}\right\|_1.$$

Then, the result in (15) is obtained using the reverse triangular inequality and the fact that

$$\left\|\boldsymbol{\mu} - \frac{\boldsymbol{\mu}_{\mathrm{B}}}{C}\right\|_1 \leq \|\boldsymbol{\mu} - \boldsymbol{\mu}_{\mathrm{B}}\|_1 + (1 - \frac{1}{C})\|\boldsymbol{\mu}_{\mathrm{B}}\|_1 \leq \|\boldsymbol{\mu} - \boldsymbol{\mu}_{\mathrm{B}}\|_1 + \|\boldsymbol{\mu}_{\mathrm{B}}\|_1$$

because $C \geq 1$.

## F    Proof of Theorem 4

If $\|\boldsymbol{\mu}^{(k)}\|_1 \leq 1/2$, we have $\mathbb{E}_{\mathrm{p}_x^*}\left(|[\hbar_1(x), \hbar_2(x), \ldots, \hbar_{t_k}(x)]\boldsymbol{\mu}^{(k)}| - 1/2\right)_+ = 0$ because $\hbar(x) \in [-1,1]$ for any $\hbar \in \mathcal{H}$. Hence, $\boldsymbol{\mu}^{(k)}$ is a $\varepsilon_{\mathrm{opt}}^{(k)}$-optimal solution of (6) with $\varepsilon_{\mathrm{opt}}^{(k)} = R^{(k)} - \overline{R}$, so that the bound in (20) is obtained as a direct consequence of the bound (34) in Theorem 7.

For the case where $\|\boldsymbol{\mu}^{(k)}\|_1 > 1/2$, let $\mathcal{F}$ be the family of functions

$$\mathcal{F} = \{f(x) = [\hbar_1(x), \hbar_2(x), \ldots, \hbar_{t_k}(x)]\boldsymbol{\mu} \text{ for some } \hbar_1, \hbar_2, \ldots, \hbar_{t_k} \in \mathcal{H}, \|\boldsymbol{\mu}\|_1 = C\}.$$

Using common properties of Rademacher complexity (see e.g., Chapter 26 in [40]), we get that the Rademacher complexity of $\mathcal{F}$ is equal to $C\mathcal{R}$. Specifically, $\mathcal{F}$ is given by convex combinations of functions in $\mathcal{H}$ scaled by $C$ because $\boldsymbol{\mu}$ in the definition of $\mathcal{F}$ can be taken to be positive since we are considering sets of base-rules $\mathcal{H}$ such that $-\hbar \in \mathcal{H}$ whenever $\hbar \in \mathcal{H}$. Hence, the family of functions

$$\mathcal{G} = \{g(x) = \left(|f(x)| - 1/2\right)_+ \text{ for some } f \in \mathcal{F}\}$$

has Rademacher complexity upper bounded by $C\mathcal{R}$, using Talagrand's contraction Lemma (see e.g., Chapter 26 in [40]) and the fact that function $h(s) = \left(|s| - 1/2\right)_+$ is $1-$Lipschitz. In addition,

$g(x) \in [0, (C - 1/2)_+]$ for any $g \in \mathcal{G}$ so that with probability at least $1 - \delta$ we have

$$\mathbb{E}_{\mathrm{P}_x^*}\left(|[\hbar_1(x), \hbar_2(x), \ldots, \hbar_{t_k}(x)]\boldsymbol{\mu}^{(k)}| - \frac{1}{2}\right)_+ \leq \frac{1}{n}\sum_{i=1}^n \left(|[\hbar_1(x_i), \hbar_2(x_i), \ldots, \hbar_{t_k}(x_i)]\boldsymbol{\mu}^{(k)}| - \frac{1}{2}\right)_+$$

$$+ 2\|\boldsymbol{\mu}^k\|_1 \mathcal{R} + \left(\|\boldsymbol{\mu}^{(k)}\|_1 - \frac{1}{2}\right)\sqrt{\frac{\log(1/\delta)}{2n}}$$

$$= 2\|\boldsymbol{\mu}^k\|_1 \mathcal{R} + \left(\|\boldsymbol{\mu}^{(k)}\|_1 - \frac{1}{2}\right)\sqrt{\frac{\log(1/\delta)}{2n}} = \varepsilon(\delta)$$

(43)

using uniform concentration bounds based on Rademacher complexity (see e.g., Chapter 3 in [30]).

Hence, $\boldsymbol{\mu}^{(k)}$ is an $\varepsilon_{\mathrm{opt}}^{(k)}$-optimal solution of (6) with $\varepsilon_{\mathrm{opt}}^{(k)} = R^{(k)} - \overline{R} + \varepsilon(\delta)$, so that the bound in (20) is obtained as a direct consequence of the bound (34) in Lemma 7 in Appendix B.

For the last result, if Algorithm 1 stops at round $k$ in Step 5, we have

$$\max_{\hbar \in \mathcal{H}} \frac{1}{n}\sum_{i=1}^n w_i \widetilde{y}_i \hbar(x_i) \leq \lambda.$$

Then, all the dual constraints are satisfied at round $k$ and we have that $R^{(k)} \leq \overline{R}$. Such inequality is obtained because $R^{(k)}$ would be the optimal value of the primal in (16) using all the base-rules, and (16) has the same objective and variables as (6) but with less constraints. Therefore, the result is obtained because the suboptimality at round $k$ satisfies $\varepsilon_{\mathrm{opt}}^k = R^{(k)} - \overline{R} + \varepsilon(\delta) \leq \varepsilon(\delta)$.

## G   Effect in Algorithm 1 of the base learner suboptimality

The next result shows how the suboptimality of solutions found by Algorithm 1 is affected by a possible early termination and the suboptimality of the base learner used in practice.

**Theorem 8.** If $\boldsymbol{\mu}^*$ is a solution of the optimization (16) using all the base-rules in $\mathcal{H}$. Then, $\boldsymbol{\mu}^{(k)}$ is an $\varepsilon_{\mathrm{op}}^{(k)}$-optimal solution of optimization (6) in Section 3 for

$$\varepsilon_{\mathrm{opt}}^{(k)} \leq \left(\max_{\hbar \in \mathcal{H}} \frac{1}{n}\sum_{i=1}^n w_i \widetilde{y}_i \hbar(x_i) - \lambda\right)_+ \|\boldsymbol{\mu}^*\|_1 + \varepsilon(\delta) \tag{44}$$

with weights $\{w_i\}$ and labels $\{\widetilde{y}_i\}$ given by (19) using the dual solution at round $k$.

*Proof.* We first show that

$$R^{(k)} \leq \overline{R} + \varepsilon_{\mathrm{base}}\|\widetilde{\boldsymbol{\mu}}\|_1 \tag{45}$$

for

$$\varepsilon_{\mathrm{base}} = \left(\max_{\hbar \in \mathcal{H}} \frac{1}{n}\sum_{i=1}^n w_i \widetilde{y}_i \hbar(x_i) - \lambda\right)_+.$$

Let $\boldsymbol{\alpha}, \boldsymbol{\beta}$ be a dual solution of the optimization problem (16) solved at round $k$. By definition of weights $\{w_i\}_{i=1}^n$ and labels $\{\widetilde{y}_i\}_{i=1}^n$ in (19), for any $\hbar \in \mathcal{H}$ we have

$$-\varepsilon_{\mathrm{base}} - \lambda \leq [\hbar(x_1), \hbar(x_2), \ldots, \hbar(x_n)](\boldsymbol{\alpha} - \boldsymbol{\beta} - \mathbf{y}/n) \leq \lambda + \varepsilon_{\mathrm{base}}.$$

In addition, the vectors $\boldsymbol{\mu}_+^* = (\boldsymbol{\mu}^*)_+, \boldsymbol{\mu}_-^* = (-\boldsymbol{\mu}^*)_+$ are feasible for the optimization problem

$$\min_{\boldsymbol{\mu}_+, \boldsymbol{\mu}_-} \frac{1}{2} - \frac{1}{n}\sum_{i=1}^n y_i \hbar(x_i)^\top (\boldsymbol{\mu}_+ - \boldsymbol{\mu}_-) + (\lambda + \varepsilon_{\mathrm{base}})\mathbf{1}^\top (\boldsymbol{\mu}_+ + \boldsymbol{\mu}_-)$$

$$\text{s.t.} -\frac{1}{2} \leq \hbar(x_i)^\top (\boldsymbol{\mu}_+ - \boldsymbol{\mu}_-) \leq \frac{1}{2}, \; i = 1, 2, \ldots, n$$

$$\boldsymbol{\mu}_+ \succeq \mathbf{0}, \boldsymbol{\mu}_- \succeq \mathbf{0}. \tag{46}$$

where $\boldsymbol{\hbar}$ is given by all the base-rules in $\mathcal{H}$. Then, using weak duality we have

$$R^{(k)} = \frac{1}{2}\Big(1 - \mathbf{1}^\top(\boldsymbol{\alpha} + \boldsymbol{\beta})\Big) \le \frac{1}{2} - \frac{1}{n}\sum_{i=1}^{n} y_i\boldsymbol{\hbar}(x_i)^\top(\boldsymbol{\mu}_+^* - \boldsymbol{\mu}_-^*) + (\lambda + \varepsilon_{\text{base}})\mathbf{1}^\top(\boldsymbol{\mu}_+^* + \boldsymbol{\mu}_-^*)$$

because $\boldsymbol{\alpha}, \boldsymbol{\beta}$ is a feasible solution of the dual of (46). Then, if $\widetilde{R}$ is the optimal value of (16) using all the base-rules in $\mathcal{H}$, we have

$$R^{(k)} \le \widetilde{R} + \varepsilon_{\text{base}}\mathbf{1}^\top(\boldsymbol{\mu}_+^* + \boldsymbol{\mu}_-^*)$$

so that the bound in (45) is obtained since $\widetilde{R} \le \overline{R}$ because $\widetilde{R}$ is the optimum value of an optimization problem with the same objective and variables as (6) but with less constraints. Therefore, the bound in (44) is obtained because

$$\varepsilon_{\text{opt}}^{(k)} = R^{(k)} - \overline{R} + \varepsilon(\delta) \le \varepsilon_{\text{base}}\|\boldsymbol{\mu}^*\| + \varepsilon(\delta).$$

$\square$

The theorem above bounds the suboptimality of RMBoost rules determined at any round by Algorithm 1. The bound in (44) accounts for the error due to the usage of relaxed constraints corresponding to the training samples through the term $\varepsilon(\delta)$. In addition, the first term in (44) accounts for the error due to a possible early termination as well as for the suboptimality in practice of the base learner used to solve (18). Notice that such suboptimality of the base learner affects any boosting method [6] and is not a significant problem for Algorithm 1 that only requires to find a violated constraint in the dual (not necessarily the most violated).

## H   Implementation details and additional experimental results

In the following we provide further implementation details and describe the datasets used in Section 6. Then, we complement the results in the main paper by including the results obtained using multiple types of label noise, assessing the running times of the methods presented, and evaluating the sensitivity to parameter $\lambda$. In the first set of additional results, we evaluate the classification performance of the proposed method in comparison with existing boosting methods in cases with uniform and symmetric label noise as well as adversarial noise; in the second set of additional results, we further show the robustness to noise of RMBoost in comparison with AdaBoost; in the third set of additional results, we compare the running times of RMBoost with AdaBoost and LPBoost; in the fourth set of additional results, we further show the performance improvement of RMBoost using large datasets; and, in the fifth set set of additional results, we show that RMBoost has little sensitivity to the choice of hyperparameter $\lambda$. In addition, the Github https://github.com/MachineLearningBCAM/RMBoost-NeurIPS-2025 provides the code of the proposed RMBoost method with the setting used in the numerical results.

### H.1   Implementation details and datasets utilized

We utilize 11 publicly available datasets that have been often use as benchmark for boosting methods: Diabetes, German Numer, Credit, Blood transfusion, Titanic, Raisin, QSAR, Climate, Susy, Higgs, and Forest covertype. These datasets can be found in the UCI repository [41] and in www.kaggle.com. The main characteristics of the datasets used is provided in Table 2.

The proposed RMBoost method is evaluated using multiple cases of label noise: the conventional symmetric and uniform label noise ($\forall x, \rho_{+1}(x) = \rho_{-1}(x) = P_{\text{noise}}$) with $P_{\text{noise}} = 10\%$ and $P_{\text{noise}} = 20\%$, and also an adversarial type of label noise with $P_{\text{noise}} = 10\%$ and $P_{\text{noise}} = 20\%$. This adversarial type of noise is implemented by flipping the labels of training instances that can be classified with high margin. Specifically, we flip the labels of the instances with the largest margins for a reference rule found with the LogitBoost method using clean labels. This type of label noise is addressed by the theoretical results presented in the paper and corresponds with non-uniform and non-symmetric noise in which $\rho_y(x) = 1$ if $y\text{h}(x)$ is large and $\rho_y(x) = 0$ otherwise, where h is the reference rule. Such type of noise describes practical situations in which an adversary chooses to change the labels in the examples that can result in the highest damage.

Table 2: Datasets characteristics

| Dataset | Samples | Instances dimensionality |
|---|---|---|
| Diabetes | 768 | 8 |
| German Numer | 1,000 | 24 |
| Credit | 690 | 15 |
| Blood transfusion | 748 | 4 |
| Titanic | 891 | 8 |
| Raisin | 900 | 7 |
| QSAR | 1,055 | 41 |
| Climate | 540 | 18 |
| Susy | 5,000,000 | 18 |
| Higgs | 11,000,000 | 21 |
| Forest covertype | 581,012 | 54 |

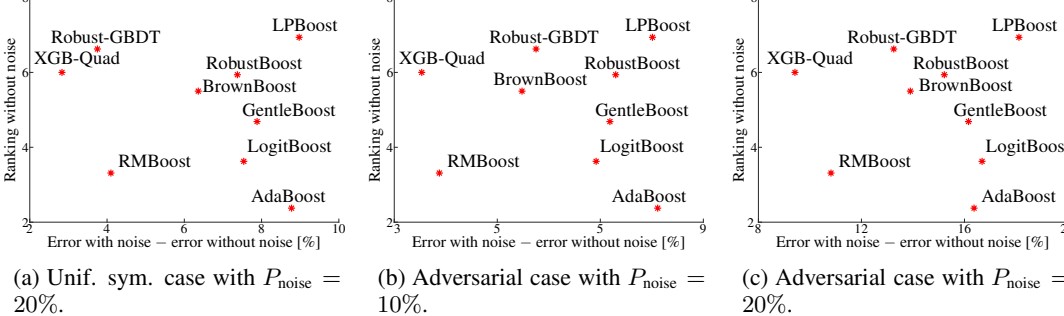

(a) Unif. sym. case with $P_{\text{noise}} = 20\%$.

(b) Adversarial case with $P_{\text{noise}} = 10\%$.

(c) Adversarial case with $P_{\text{noise}} = 20\%$.

Figure 3: Trade-off classification performance vs robustness to uniform and symmetric noise as well as adversarial noise.

The proposed RMBoost method is compared with 8 boosting methods: the 4 state-of-the-art techniques AdaBoost, LogitBoost, GentleBoost, and LPBoost together with the 4 robust methods RobustBoost, BrownBoost, XGBoost with quadratic potential (XGB-Quad), and Robust-GBDT, specifically designed for scenarios with noisy labels. RMBoost and all the methods used for comparison are implemented using default values for hyper-parameters and the code in standard libraries or provided by the authors. Methods RobustBoost, AdaBoost, LogitBoost, GentleBoost, and LPBoost are implemented using their Matlab codes, methods XGB-Quad and BrownBoost are implemented using the Python libraries 'XGBoost' `https://xgboost.readthedocs.io` and 'BrownBoost' `https://github.com/lapis-zero09/BrownBoost`, respectively, and method Robust-GBDT is implemented using the code provided by the authors [38]. The proposed RMBoost is implemented by learning parameters $\boldsymbol{\mu}^* \in \mathbb{R}^t$ and base-rules $\hbar_1, \hbar_2, \ldots, \hbar_t$ using Algorithm 1, and by predicting labels using the deterministic classifier $\mathrm{h}^{\mathrm{d}}_{\boldsymbol{\mu}^*}(x) = \mathrm{sign}([\hbar_1(x), \hbar_2(x), \ldots, \hbar_t(x)]\boldsymbol{\mu}^*)$. In particular, we use simplex-based solvers for linear optimization with tolerances for constraints and dual feasibility of $10^{-3}$, and we take $\lambda = 1/\sqrt{n}$ in all the numerical results.

## H.2 Additional experimental results

In the first set of additional experimental results, we further compare the classification error of RMBoost with existing boosting methods. The results in Table 1 in the paper as well as Table 3 below are obtained carrying out 100 random and stratified train/test partitions with 10% test samples. Table 1 in the paper shows the classification error obtained by the most representative methods with clean labels and with symmetric and uniform label noise with $P_{\text{noise}} = 10\%$. Table 3 shows the classification error obtained by the 9 boosting methods with clean labels, with symmetric and uniform label noise with $P_{\text{noise}} = 10\%$ and $P_{\text{noise}} = 20\%$, as well as with adversarial noise with $P_{\text{noise}} = 10\%$ and $P_{\text{noise}} = 20\%$. Over multiple datasets and types of label noise, Table 1 together with Table 3 show that the minimax risks obtained at learning are often near the prediction error and that RMBoost can obtain top accuracies in comparison with existing boosting methods both in noise-less and noisy cases.

Table 3: Average classification error in % $\pm$ st. dev. for RMBoost and state-of-the-art boosting methods.

| | Method | Titanic | German | Blood | Credit | Diabetes | Raisin | QSAR | Climate |
|---|---|---|---|---|---|---|---|---|---|
| **Noise-less** | RobustBoost | 21±3.7 | 25±4.6 | 26±4.4 | 15±4.7 | 28±5.6 | 16±3.4 | 16±3.5 | 9.1±2.9 |
| | AdaBoost | 20±3.2 | 24±4.2 | 24±4.4 | 14±3.9 | 27±4.9 | 15±2.7 | 14±3.1 | 8.5±2.0 |
| | LPBoost | 32±5.7 | 28±2.4 | 32±5.8 | 16±4.3 | 29±5.5 | 16±3.2 | 16±3.1 | 8.3±2.0 |
| | LogitBoost | 21±3.7 | 24±4.5 | 27±4.3 | 14±4.0 | 26±5.3 | 15±2.6 | 14±3.1 | 8.5±2.0 |
| | GentleBoost | 22±3.7 | 25±4.6 | 26±4.1 | 14±4.2 | 27±5.2 | 15±2.9 | 14±3.1 | 8.7±2.3 |
| | BrownBoost | 20±3.7 | 25±3.6 | 25±3.3 | 15±4.1 | 34±5.0 | 15±3.9 | 14±3.4 | 11±2.4 |
| | Robust-GBDT | 23±3.9 | 24±3.4 | 24±3.6 | 23±4.8 | 33±4.4 | 16±3.6 | 16±3.8 | 11±2.7 |
| | XGB-Quad | 21±3.7 | 25±3.4 | 22±3.9 | 22±5.6 | 34±4.6 | 16±3.8 | 23±3.5 | 8.4±2.0 |
| | RMBoost | 22±3.5 | 27±2.8 | 20±5.4 | 14±5.6 | 26±4.5 | 12±3.6 | 15±3.1 | 7.5±2.0 |
| | Minimax risk | 20±0.3 | 26±0.6 | 24±0.6 | 16±0.4 | 25±0.8 | 14±0.6 | 17±0.5 | 9.3±0.4 |
| **$P_{\text{noise}} = 10\%$** | RobustBoost | 21±4.2 | 29±3.9 | 26±4.2 | 19±4.5 | 31±5.3 | 20±3.9 | 19±4.0 | 15±4.2 |
| | AdaBoost | 22±4.1 | 30±4.2 | 27±3.9 | 18±4.5 | 31±5.2 | 19±3.9 | 19±3.5 | 12±2.8 |
| | LPBoost | 35±6.1 | 34±5.4 | 36±5.7 | 22±4.7 | 32±5.7 | 21±4.1 | 20±3.5 | 12±3.6 |
| | LogitBoost | 23±4.5 | 29±4.4 | 28±4.3 | 19±4.5 | 29±5.1 | 19±4.1 | 18±3.5 | 10±2.9 |
| | GentleBoost | 24±4.5 | 29±4.2 | 29±4.3 | 19±4.7 | 30±4.9 | 19±4.0 | 18±3.5 | 10±2.9 |
| | BrownBoost | 23±3.9 | 28±3.8 | 26±4.4 | 21±4.4 | 35±4.9 | 17±3.9 | 18±3.8 | 11±2.8 |
| | Robust-GBDT | 24±3.9 | 25±3.3 | 25±3.2 | 26±5.6 | 34±4.1 | 20±4.1 | 19±3.5 | 20±2.9 |
| | XGB-Quad | 22±3.9 | 27±4.1 | 23±3.4 | 24±4.8 | 34±4.5 | 20±3.4 | 26±4.7 | 10±3.2 |
| | RMBoost | 22±3.6 | 27±5.0 | 22±3.9 | 16±3.8 | 27±5.1 | 14±2.4 | 20±3.3 | 9.5±2.8 |
| | Minimax risk | 24±0.8 | 29±0.8 | 28±0.9 | 21±0.8 | 28±1.1 | 19±1.0 | 23±0.8 | 15±0.8 |
| **$P_{\text{noise}} = 20\%$** | RobustBoost | 24±4.0 | 33±5.1 | 28±4.8 | 25±5.0 | 34±5.6 | 25±4.9 | 25±4.3 | 20±5.9 |
| | AdaBoost | 25±4.2 | 34±5.0 | 30±4.9 | 24±5.3 | 34±4.8 | 24±4.7 | 24±4.0 | 20±3.9 |
| | LPBoost | 39±6.4 | 37±4.8 | 40±6.0 | 28±5.2 | 36±5.4 | 27±2.2 | 26±4.3 | 17±5.4 |
| | LogitBoost | 28±4.4 | 33±4.8 | 32±4.7 | 24±6.0 | 32±5.3 | 23±4.6 | 24±3.8 | 14±4.2 |
| | GentleBoost | 28±4.3 | 34±5.2 | 32±4.8 | 25±5.2 | 34±5.0 | 25±5.2 | 24±3.8 | 14±4.2 |
| | BrownBoost | 28±4.8 | 31±4.6 | 30±5.0 | 25±4.8 | 38±5.3 | 21±4.0 | 23±4.4 | 15±4.1 |
| | Robust-GBDT | 22±4.6 | 27±2.8 | 27±3.5 | 18±5.1 | 31±4.3 | 19±4.5 | 19±3.6 | 12±6.0 |
| | XGB-Quad | 23±4.6 | 29±4.2 | 24±3.7 | 26±6.1 | 35±5.0 | 20±4.1 | 26±4.8 | 11±3.8 |
| | RMBoost | 25±4.2 | 29±2.3 | 27±4.1 | 16±4.0 | 28±6.3 | 18±1.9 | 24±3.3 | 20±3.7 |
| | Minimax risk | 30±1.1 | 32±0.8 | 33±0.8 | 27±0.9 | 30±1.4 | 25±1.1 | 27±0.6 | 20±1.1 |
| **Adversarial $P_{\text{noise}} = 10\%$** | RobustBoost | 26±4.0 | 34±4.6 | 31±5.3 | 22±5.1 | 37±5.8 | 23±4.2 | 23±4.3 | 17±4.6 |
| | AdaBoost | 26±3.9 | 34±5.0 | 32±4.9 | 22±4.8 | 36±6.2 | 23±3.8 | 24±3.6 | 14±3.2 |
| | LPBoost | 38±5.7 | 38±4.6 | 40±5.8 | 25±5.5 | 38±5.6 | 24±4.4 | 25±4.1 | 13±3.5 |
| | LogitBoost | 28±3.5 | 33±5.2 | 33±5.2 | 21±4.3 | 34±5.0 | 23±4.0 | 22±3.8 | 11±3.2 |
| | GentleBoost | 28±3.9 | 33±4.7 | 34±5.1 | 22±4.6 | 35±5.1 | 23±3.7 | 22±3.7 | 11±3.1 |
| | BrownBoost | 29±4.3 | 33±4.3 | 30±5.3 | 20±4.4 | 36±5.7 | 21±4.0 | 22±3.6 | 13±3.1 |
| | Robust-GBDT | 28±3.8 | 31±3.9 | 28±4.1 | 23±4.7 | 31±4.8 | 25±4.4 | 22±3.4 | 24±6.2 |
| | XGB-Quad | 25±4.5 | 30±5.1 | 26±3.2 | 17±4.8 | 29±4.7 | 19±3.8 | 24±5.0 | 10±3.1 |
| | RMBoost | 25±5.4 | 27±3.9 | 26±3.1 | 19±5.1 | 22±3.6 | 16±3.7 | 23±3.8 | 11±2.2 |
| | Minimax risk | 25±0.9 | 33±0.5 | 28±0.7 | 25±0.7 | 29±0.5 | 21±1.3 | 24±0.3 | 17±0.5 |
| **Adversarial $P_{\text{noise}} = 20\%$** | RobustBoost | 30±4.6 | 44±5.6 | 36±5.7 | 33±5.2 | 46±5.6 | 33±4.1 | 26±4.7 | 30±5.9 |
| | AdaBoost | 30±4.2 | 43±5.3 | 37±5.7 | 32±5.4 | 46±6.0 | 33±4.3 | 27±4.3 | 30±5.2 |
| | LPBoost | 42±5.7 | 46±4.9 | 46±6.8 | 36±6.9 | 48±5.6 | 35±4.6 | 25±4.8 | 42±5.6 |
| | LogitBoost | 33±4.5 | 43±5.2 | 39±5.9 | 32±5.6 | 45±5.8 | 33±4.1 | 22±4.5 | 33±5.2 |
| | GentleBoost | 33±4.6 | 42±5.6 | 39±6.2 | 32±5.7 | 46±5.7 | 33±4.4 | 22±4.7 | 33±6.0 |
| | BrownBoost | 33±4.6 | 43±5.4 | 33±5.4 | 29±5.0 | 46±5.7 | 31±4.0 | 22±4.8 | 33±5.2 |
| | Robust-GBDT | 32±3.7 | 37±4.1 | 33±4.9 | 30±5.0 | 41±5.0 | 32±4.3 | 38±5.0 | 32±6.3 |
| | XGB-Quad | 29±5.0 | 42±5.2 | 29±5.4 | 22±5.6 | 38±6.5 | 30±5.7 | 20±4.6 | 29±5.8 |
| | RMBoost | 27±5.4 | 31±3.4 | 34±5.8 | 20±3.6 | 29±3.3 | 23±5.5 | 22±4.8 | 27±5.5 |
| | Minimax risk | 28±0.9 | 35±0.3 | 33±0.9 | 29±0.6 | 31±0.6 | 27±1.6 | 19±0.6 | 28±0.7 |

Figure 1 in the paper as well as Figure 3 and Table 4 summarize the results in Tables 1 and 3 in terms of the trade-off between classification performance and robustness to noise. The vertical axis of the figures shows the classification performance in terms of the average ranking in the noise-less case, while the horizontal axis shows the robustness to noise in terms of the average difference between the error with noisy labels and that without noise. Figure 3 and Table 4 extend the results in the main paper to uniform and symmetric label noise with $P_{\text{noise}} = 20\%$ and with adversarial noise with $P_{\text{noise}} = 10\%$ and $P_{\text{noise}} = 20\%$ complementing those with uniform and symmetric label noise with $P_{\text{noise}} = 10\%$ in Table 1 and Figure 1. Over multiple types of label noise, Figures 1 and 3 together with Table 4 show that RMBoost is a robust method that can also provide a strong classification performance near that of AdaBoost method.

In the second set of additional results, we further show the robustness to noise of RMBoost in comparison with AdaBoost. Figure 2 in the paper as well as Figures 4a and 4b are obtained computing for each noise level the classification error over 500 random stratified partitions with 10% test samples. Figures 4a and 4b extend the results using 'Diabetes' and 'Climate' datasets completing those in the main paper that show the results using Credit dataset. Figures 4a and 4b show similar behavior to Figure 2 in the paper. In particular, the figures show that RMBoost method is significantly less

Table 4: Classification performance and robustness to noise for RMBoost and state-of-the-art boosting methods.

| Method | Ranking without noise | Error with noise - error without noise [%] | | | |
|---|---|---|---|---|---|
| | Average rank | Noise 10% | Noise 20% | Adver. noise 10% | Adver. noise 20% |
| RobustBoost | 5.94 | 3.08 | 7.37 | 7.30 | 15.22 |
| AdaBoost | 2.38 | 3.88 | 8.78 | 8.11 | 16.37 |
| LPBoost | 6.94 | 4.39 | 8.97 | 8.01 | 18.12 |
| LogitBoost | 3.63 | 3.27 | 7.54 | 6.92 | 16.68 |
| GentleBoost | 4.69 | 3.38 | 7.88 | 7.19 | 16.16 |
| BrownBoost | 5.50 | 2.72 | 6.36 | 5.48 | 13.90 |
| Robust-GBDT | 6.63 | 2.82 | 3.75 | 5.75 | 13.26 |
| XGB-Quad | 6.00 | 1.65 | 2.83 | 3.53 | 9.42 |
| RMBoost | 3.31 | 1.71 | 4.10 | 3.88 | 10.82 |

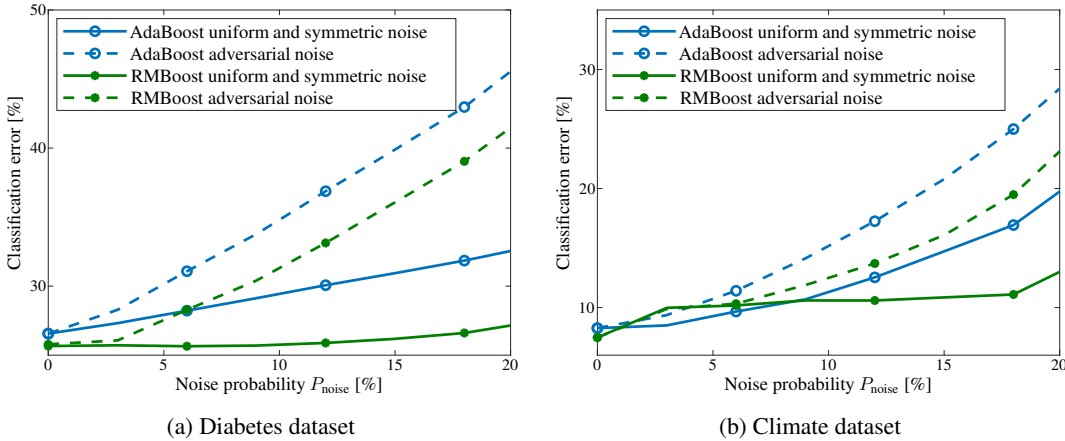

(a) Diabetes dataset        (b) Climate dataset

Figure 4: Performance degradation of AdaBoost and RMBoost methods for increased levels of noise.

affected by increased levels of noise. In particular, RMBoost performance only mildly deteriorates with label noise, in accordance with the theoretical results shown in the paper.

## H.3 Comparison in terms of running times

In the third set of additional results, we compare the running times of RMBoost with those of AdaBoost and LPBoost. Figure 5 shows the relative running times of the methods varying the training sizes using 'Credit' and 'QSAR' datasets (the absolute running times in all the methods are in the order of seconds in a regular desktop machine). The vertical axis in the figure represents the ratio between the learning running times for different training sizes divided by that achieved with 100 training samples averaged over 100 random partitions. In accordance with the discussion in Sec-

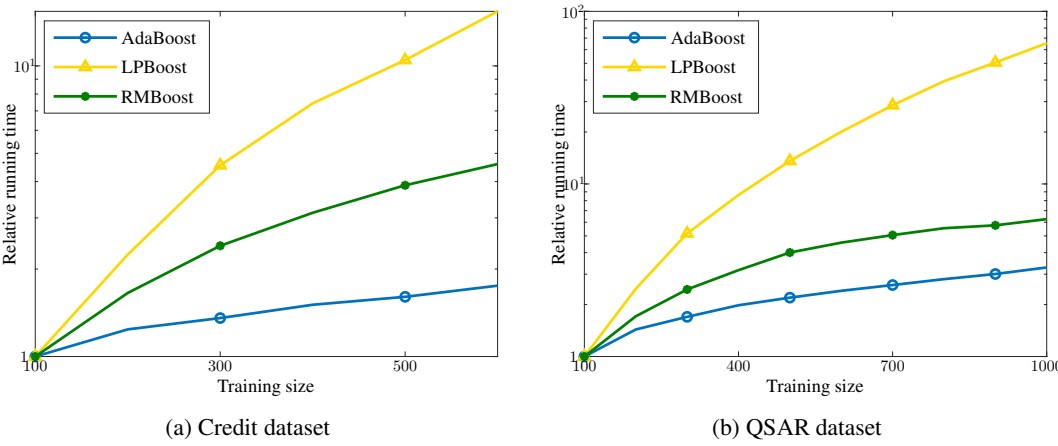

(a) Credit dataset        (b) QSAR dataset

Figure 5: Comparison of relative running times vs training sizes for RMBoost, LPBoost, and AdaBoost.

Table 5: Average classification error in % ± st. dev. for RMBoost and state-of-the-art boosting methods.

| | Dataset | RobustB | AdaB | LPB | LogitB | GentleB | BrownB | GBDT | XGB-Q | RMB | Minimax |
|---|---|---|---|---|---|---|---|---|---|---|---|
| Noiseless | Susy | 24±1.7 | 24±1.8 | 30±2.2 | 24±2.0 | 25±1.7 | 23±1.7 | 25±1.8 | 24±1.6 | 23±2.0 | 23±0.3 |
| | Higgs | 34±1.8 | 33±2.0 | 38±3.1 | 33±1.9 | 34±2.1 | 34±1.9 | 37±2.0 | 37±2.4 | 35±2.1 | 33±0.3 |
| | Forestcov | 20±1.8 | 20±1.8 | 27±1.7 | 17±1.1 | 20±1.7 | 20±1.5 | 26±2.0 | 33±1.7 | 22±1.6 | 22±0.2 |
| 10% | Susy | 26±2.2 | 24±1.8 | 35±3.4 | 27±1.9 | 28±1.8 | 24±1.8 | 26±1.8 | 24±1.8 | 23±1.9 | 28±0.4 |
| | Higgs | 35±2.1 | 35±1.9 | 41±1.4 | 36±2.1 | 37±2.3 | 35±2.2 | 39±2.2 | 38±2.4 | 35±2.4 | 36±0.5 |
| | Forestcov | 22±1.7 | 22±1.7 | 34±1.9 | 22±1.2 | 25±2.0 | 23±1.6 | 27±2.1 | 33±2.2 | 23±1.7 | 28±0.4 |
| 20% | Susy | 27±2.0 | 26±2.0 | 38±2.2 | 30±2.1 | 32±2.0 | 25±1.9 | 29±1.6 | 25±1.7 | 24±2.0 | 32±0.5 |
| | Higgs | 37±2.3 | 37±2.0 | 46±3.3 | 38±2.3 | 39±2.4 | 37±2.3 | 41±2.9 | 39±2.7 | 36±2.2 | 39±0.6 |
| | Forestcov | 24±1.8 | 24±1.8 | 38±2.3 | 25±1.5 | 29±1.9 | 25±1.9 | 32±2.1 | 34±2.7 | 23±2.0 | 32±0.4 |
| Adv 10% | Susy | 32±2.0 | 32±2.1 | 43±2.7 | 34±2.0 | 35±2.0 | 31±3.3 | 34±1.6 | 33±2.9 | 32±2.3 | 28±0.5 |
| | Higgs | 39±2.0 | 39±2.0 | 48±2.8 | 41±2.0 | 42±2.3 | 38±3.4 | 44±1.2 | 38±1.7 | 38±2.4 | 40±0.5 |
| | Forestcov | 28±1.9 | 28±2.0 | 40±2.2 | 28±2.1 | 29±2.3 | 28±2.0 | 27±1.7 | 28±2.3 | 28±2.1 | 31±0.5 |
| Adv 20% | Susy | 40±2.0 | 40±1.9 | 49±2.8 | 43±1.9 | 44±2.2 | 40±1.5 | 41±2.3 | 39±2.0 | 38±2.5 | 33±0.5 |
| | Higgs | 49±2.1 | 50±2.2 | 49±3.4 | 49±2.7 | 49±2.4 | 49±2.8 | 49±0.7 | 49±4.1 | 48±3.3 | 46±0.6 |
| | Forestcov | 39±2.5 | 39±2.4 | 47±3.8 | 38±2.2 | 40±2.2 | 39±2.1 | 38±1.4 | 36±2.7 | 36±2.6 | 36±0.7 |

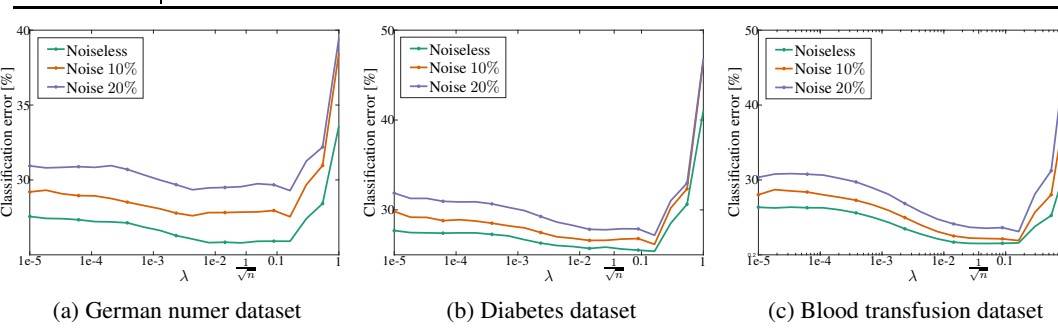

(a) German numer dataset     (b) Diabetes dataset     (c) Blood transfusion dataset

Figure 6: Performance of RMBoost method using multiple values of $\lambda$.

tion 5 of the main paper, the results depicted in the figure show that RMBoost can achieve similar running times as LPBoost method, which also addresses a linear optimization problem at learning. As expected, AdaBoost method results in lower running times since it does not require to solve an optimization problem at each round. Nevertheless, the complexity increase required by RMBoost is not significant and scales mildly with the training size.

### H.4 Additional results with larger datasets

In the fourth set of additional results, we further compare the classification error of RMBoost with existing boosting methods using large datasets. Table 1 in the paper and Table 3 in Appendix H.2 shows the classification error obtained by using small datasets (up to 1000 samples). The Table 5 shows the classification error obtained by using 5,000 randomly drawn training samples from the 'Susy,' 'Higgs,' and 'Forest Covertype' datasets. Such table shows the results with clean labels, with symmetric and uniform label noise with $P_{\text{noise}} = 10\%$ and $P_{\text{noise}} = 20\%$, as well as with adversarial noise with $P_{\text{noise}} = 10\%$ and $P_{\text{noise}} = 20\%$. The additional results in Table 5 show similar behavior as those in Tables 1 and 3 using small datasets. The proposed methods achieve adequate performance without noise together with improved robustness with noisy labels.

### H.5 Hyperparameter sensitivity

In the fifth set of additional results, we asses the sensitivity of the RMBoost method to the choice of hyperparameter $\lambda$. These numerical results are obtained computing for each noise level the classification error over 200 random stratified partitions with 10% test samples. Figure 6 shows the classification error in 3 datasets obtained by varying the hyperparameter $\lambda$ in cases without label noise and with 10% and 20% uniform and symmetric noise. The figure shows that the performance is not significantly affected by the choice of hyperparameter $\lambda$. Although better results can be obtained by tuning the value of $\lambda$, the default value of $1/\sqrt{n}$ achieves adequate results in general.

