# OpenReview forum: "Robust Minimax Boosting with Performance Guarantees"
_NeurIPS.cc/2025/Conference — NeurIPS 2025 poster_

### Official Review · Reviewer_YmwY · 2025-06-22

**Clarity:** 3
**Significance:** 3
**Originality:** 3
**Rating:** 5
**Confidence:** 4

**Summary:**

The paper introduces RMBoost, a boosting method that directly minimizes the worst-case error probability over an uncertainty set of distributions. This provides a principled approach to achieving robustness against label noise, moving beyond the use of specific surrogate loss functions. The key contributions are: (1) A novel minimax formulation for robust boosting that is shown to be equivalent to a tractable convex optimization problem. (2) Finite-sample performance guarantees that, unlike much prior work, holds for general types of label noise and explicitly characterizes the impact of noise properties.

**Questions:**

1. The minimax risk R computed by the algorithm is shown to be a good estimate of the final classification error (Table 1). Could this value be used in a practical way, for example, for model selection? For instance, could one use R to help tune the hyperparameter λ without a separate validation set?

2. The margin constraints in the optimization problem (6), prevent any single data point from having an arbitrarily large positive or negative margin, which intuitively explains the robustness. How does this compare to other regularization techniques in boosting, such as explicitly penalizing the variance of the margins as in [23, 24]?

**Ethical Concerns:**

["NO or VERY MINOR ethics concerns only"]

**Final Justification:**

I thank the authors for their replies to my questions. I maintain my support of this work.

**Limitations:**

nothing to add

**Paper Formatting Concerns:**

The format is fine.

**Quality:**

3

**Strengths And Weaknesses:**

Strengths:

Principled and Novel Formulation: The core contribution—framing robust boosting as a distributionally robust optimization problem (Eq. 4-5)—is elegant and powerful. It sidesteps the often heuristic choice of robust potential functions (e.g., unhinged, sigmoid) and instead tackles the problem from first principles by directly minimizing the worst-case 0-1 loss. Showing this is equivalent to a tractable linear program (Theorem 1) makes the approach practical.

Strong and General Theoretical Guarantees: The paper provides significant theoretical results. The finite-sample generalization bound (Theorem 2) is a key strength. It holds for general label noise, going beyond the common but restrictive assumptions of symmetric or uniform noise. The bound's explicit dependence on the noise probability (Pnoise) and variance (Varp∗{ρy(x)}) offers a much clearer picture of how different noise characteristics affect performance. Furthermore, the proof of Bayes consistency (Theorem 3) establishes the method's soundness in the limit of expressive base-learners.

Efficient and Scalable Algorithm: The proposed learning algorithm (Algorithm 1), based on column generation, is a standard and well-suited technique for this type of problem (similar to LPBoost). This demonstrates that the theoretically-principled formulation does not come at the cost of practicality. The analysis of the algorithm's convergence and optimality (Theorem 4) is also a valuable contribution.
Weaknesses

The weaknesses of this paper are minor and do not detract from the overall quality of the work.

Motivation for Hyperparameter Choice (λ): The performance of the method depends on the radius of the uncertainty set, λ. The authors propose a default value of λ=1/\sqrt(n), which is well-motivated by standard concentration inequalities. While the sensitivity analysis in Appendix H (Figure 6) is excellent and shows that the method is not overly sensitive to this choice, the main text could benefit from a slightly deeper discussion. For instance, briefly mentioning that this parameter could be tuned via cross-validation and explaining the trade-off it controls (a larger λ implies more robustness but potentially more bias) would be helpful for the reader.

Clarity on the Adversarial Noise Model: The adversarial noise model is a good way to test a worst-case scenario. However, its definition relies on a "reference rule found with the LogitBoost method using clean labels" (Appendix H, line 611). This implies an oracle with access to the true labels is needed to construct the noisy dataset for the simulation. This is perfectly acceptable for an experimental setup, but it should be stated with perfect clarity in the main text that this is a simulated worst-case attack designed to probe the algorithm's limits, rather than a naturally occurring noise process, to prevent any misunderstanding.

---

> ### Author Rebuttal · Authors · 2025-07-30
>
> We would like to thank the Reviewer for his/her appreciation of the principled approach proposed to achieve robustness against label noise. We also thank the Reviewer for the suggestions and comments provided that are addressed below. We will be happy to answer any further question the Reviewer could have during the authors/reviewers discussion period.
>
> **Extended discussion for the hyper-parameter $\lambda$**
>
> Following the Reviewer’s suggestion, we plan to use the extra page allowed in the final version to expand the discussion about the hyper-parameter $\lambda$, as follows. The hyper-parameter $\lambda>0$ accounts for the error in the finite-sample average in (5) and can be selected using standard cross-validation approaches. This selection can be enhanced taking into account the family of base-rules used or prior knowledge on the amount of label noise. In particular, more complex families of base-rules or increased levels of noise can benefit from higher values for $\lambda$. A simple default value for such parameter is $\lambda=1/\sqrt{n}$, which is the value used in all the experimental results in the paper (Appendix H further analyzes the sensitivity of the proposed methods to the choice of that hyperparameter). Overall, the hyperparameter $\lambda$ controls the method’s robustness to underlying distributions that depart from the empirical distribution of training samples. In particular, larger values for $\lambda$ can result in increased robustness (larger uncertainty set) but too large values for $\lambda$ can result in minimax rules that are too conservative since the uncertainty set would contain quite adversarial distributions.
>
> **Rationale for the adversarial noise model used in the experimental results**
>
> As described by the Reviewer, the type of adversarial noise model used in the experimental results does not occur naturally in cases where the noise has a stochastic origin. Such type of noise can be thought of as an extreme case used in the experimental results to test the robustness potential of the methods proposed. In addition, such type of noise can be thought of as a type of label corruption that could be injected in a dataset by an adversary. Specifically, an adversary could have access to a clean dataset and modify the labels that the adversary considers could harm the most the learning algorithm. We will clarify this rationale for the adversarial noise used in the experimental results, to avoid possible confusion.
>
> **Usage of the minimax risk $R$ for model selection**
>
> This is a very interesting question from the Reviewer. Indeed, we believe that using that value for model selection can be quite appropriate, and somehow similar to the approach proposed in [37] to select the family of base-rule considered using performance bounds. Specifically, the hyper-parameters of the trees used (e.g., maximum number of terminal nodes, maximal depth, and minimum number of leaf samples) or the number of boosting rounds could be selected using the minimax risk $R$ without requiring a separate validation set. However, the value of the minimax risk $R$ is not very adequate to select the hyper-parameter $\lambda$ since, for any dataset, larger values for $\lambda$ will result in larger values for the minimax risk $R$, as the uncertainty set in (5) increases with $\lambda$.
>
> **Relation between the margin constraints in optimization problem (6) and variance penalization in other methods**
>
> Both ideas are algorithmically quite different but are somehow related in the sense that both approaches can achieve an increased distribution of margins. An average margin near its maximum value (1/2) together with the margin constraints in (6) force all the margins to be high (near 1/2). Similarly, a large average margin together with a small margin variance would also result in increased distribution of margins. However, the optimization problem resulting in cases with variance penalization is quite different to that in (6). In particular, the maximization of the average margin coupled with a minimization of the margins’ variance is a multiobjective problem that is addressed in existing works by using a hyperparameter that balances the relevance of both objectives. In addition, the subsequent optimization problem is quadratic due to the variance term and, to the best of our knowledge, there are no theoretical results describing the robustness to label noise of methods based on variance penalization.

---

> > ### Comment · Reviewer_YmwY · 2025-08-05
> >
> > I thank the authors for their replies to my questions. I maintain my support of this work.

---

### Official Review · Reviewer_5GJn · 2025-07-01

**Clarity:** 1
**Significance:** 3
**Originality:** 3
**Rating:** 4
**Confidence:** 1

**Summary:**

This paper formulates a min-max optimization method over uncertainty sets (max), finding a sequence of base learners minimizing the loss. This seems analogous to DRO, though the authors only give brief mention to this. The authors present several analytical results, and a convincing set of empirical results over many baseline models and datasets.

**Questions:**

1. Could you please reformulate both Eq 5 and 7 s.t. I can understand the intuition of these problems?

2. Could you also present Algorithm 1 in more abstraction, so that someone might understand it outside of the dense notation?

**Ethical Concerns:**

["NO or VERY MINOR ethics concerns only"]

**Final Justification:**

The authors present significant analytical results for the proposed minimax boosting procedure.

**Limitations:**

The authors don't address this issue. The authors state "The limitations are discussed over the paper. In particular the theorems state
697 the hypothesis required." but this misses the above intent and makes finding relevant text 'throughout' very difficult. Presumably, the authors state about the scope of their problem: if the assumptions or hypotheses don't hold then their method might fail (i.e. every method's most general limitation).

**Quality:**

2

**Strengths And Weaknesses:**

Strengths

1. Overall, the authors present a rigorous set of analytical results on a novel model. (Presumably) the proofs seem to be of high quality and require a good deal of analysis in the appendix lemmas.

2. The method seems to evaluate quite well. The authors primary results demonstrate consistent quantitative results (Table 1), and some simple qualitative and runtime results (in Appendix). So, the method does seem to have strong, practical value.

Weaknesses:

1. Overall, this was an extremely difficult paper to interpret. While I am familiar with both ERM and DRO works, I found the presentation to be very difficult to translate to those works. The notation is very terse, and the authors don't elaborate on base-rules, or the uncertainty set in (5). While I understand this refers to a "uncertainty set," I don't actually understand the intuition of (5): 140-144 define the terms but don't give a high level statement outside of notation.

Since this is a primary element in the minmax formulation (defined after the problem formulation in (4)), I have to reduce my confidence to 1, and adjust the quality and clarity. I further could not interpret the Thm 1 or its proof (including its lemmas)

2. Similar to 1, even the algorithm description is impenetrable to me. This is primarily a wall of symbols with insufficient description. One could scarcely implement this method from this algorithm box.  The authors only cite one subproblem (16). Further use of this might mitigate the presentation complexity. Overall, from Algorithm 1 I couldn't describe the method in any more technical terms than Eq (4).

3. The authors could improve their related work to better taxonomize the gap they are addressing. The authors mix some of the "background" notation, which takes away from the presentation of the gap. This really muddles the primary contributions (which are presented in L44, but are less specific in the prior work)

---

> ### Author Rebuttal · Authors · 2025-07-30
>
> We thank the Reviewer for the suggestions provided regarding notations and more intuitive descriptions. We trust the explanations provided below can help the Reviewer to get a more clear interpretation of the results presented and the relation with the related work. In case there are still some notational barriers limiting the results clarity, we would appreciate if the Reviewer let us know what specific notations are unclear so that we could further elaborate in the updated manuscript.
>
> **Notation clarification and intuition for uncertainty sets and Theorem 1**
>
> We plan to use the extra page allowed in the final version of the paper to further clarify the notation used with a more intuitive description. Regarding base-rules, the theoretical sections of the paper consider general families of base-rules (any family of bounded functions from $\mathcal{X}$ to $[-1,1]$). In some parts of the main text as lines 246-248, 321-324, and in the experimental results, the family of base-rules is given by decision trees, which are the common type of base-rules used in boosting methods.
>
> The uncertainty set in (5) is composed by all the probability distributions over instance-label pairs for which expectations of the base-rules are near their empirical averages over the training samples. Specifically, the expectations used to define the uncertainty set are correlations between labels and base-rules predictions ($y \boldsymbol{\hbar}(x)$). The rationale for the usage of such uncertainty set is to have a small set that can contain the true underlying distribution of samples with high probability. The uncertainty set in (5) is small because the family of base rules is often very large so that (5) is given by a large number of constraints. The uncertainty set in (5) contains the true underlying distribution with high probability because base-rules are simple functions for which uniform convergence of sample averages is ensured.
>
> Theorem 1 shows that the minimax problem in (4) is equivalent to a tractable convex optimization problem (linear optimization with L1-regularization). This result enables to address (4) in practice by solving the tractable problem (6). The main idea behind the proof of Theorem 1 is substituting the inner maximization in (4) by its Fenchel dual, so that the minimax problem becomes a convex optimization problem.
>
> **More intuitive formulation of Equations (5) and (7)**
>
> Regarding equation (5) for the uncertainty set, other way to formulate the same uncertainty set is as follows. Let $\mathcal{H} =\lbrace h_1, h_2, \dots, h_T \rbrace $ be the set of all base-rules, e.g., all the decision trees with a bounded number of nodes given by components of the instances in the training set. The uncertainty set in (4) is
> $$\mathcal{U}=\lbrace\text{prob. dist. }\mathrm{p}\text{ over }\mathcal{X}\times \mathcal{Y}\text{ such that }|\mathbb{E}_p\lbrace y h_j(x)\rbrace -\frac{1}{n} \sum\_{i=1}^n y_i h_j(x_i)|\leq \lambda,\text{ for all }j=1,2,…T\rbrace$$ that is, $\mathcal{U}$ is given by the $2T$ linear constraints  $$\frac{1}{n}\sum\_{i=1}^n y_i h_j(x_i)-\lambda\leq \sum\_{x,y} p(x,y) yh_j(x)\leq \frac{1}{n}\sum\_{i=1}^n y_i h_j(x_i)+\lambda,\  \mbox{ for } j=1,2,\ldots,T$$
> The Lagrange (Fenchel) multipliers of such constraints correspond to the parameters $\boldsymbol{\mu}$ in Theorem 1.
>
> Regarding equation (7) for the minimax classification rule, other way to formulate the equation defining the classification rule is as follows. Let $\boldsymbol{\mu}^{\*}=[{\mu}^{\*}\_1, {\mu}^{\*}\_2,…, {\mu}^{\*}\_T]^{\dagger}$ be the solution of optimization problem (7).
>
> A solution of the minimax problem in (4) is given by
> $$h\_{\boldsymbol{\mu}^{\*}}(y|x)=y \sum\_{j=1}^T h_j(x){\mu}^{\*}\_j +1/2$$
> that is, the minimax classification rule is given by a linear combination of the base-rules $\mathcal{H}=\lbrace h_1,h_2,…, h_T\rbrace$ with coefficients given by a solution of the optimization problem (6). For instances $x$ for which the combination of base-rules $\sum\_{j=1}^T h\_j(x)\mu^{\*}\_j$ is positive, the probability $h_{\boldsymbol{\mu}^{\*}}(y=+1|x) $ is larger than 1/2 and hence larger than $h_{\boldsymbol{\mu}^{\*}}(y=-1|x)$, so that the deterministic classifier predicts label +1, (otherwise the classifier predicts label -1). Therefore, the determinist minimax classifier is given by
> $$h_{\boldsymbol{\mu}^{\*}}(x)=\text{sign}(\sum\_{j=1}^T h_j(x)\mu^{\*}_j)$$
> as shown in equation (8).
>
> **Abstract description of Algorithm 1**
>
> Algorithm 1 follows a similar approach to other boosting methods based on column generation [25,34-36]. In each boosting round, a base learner finds the new base-rule (e.g., decision tree) by finding a base-rule that best fits a set of weighted samples ${(x_i,\tilde{y}_i,w_i)}$ obtained from the training samples (line 3 of the Algorithm). Then, the coefficients for the current set of selected base-rules are obtained by solving the linear optimization problem (16) (line 7 of the Algorithm) and the dual solution of such optimization is used to obtain the weighted samples ${(x_i,\tilde{y}_i,w_i)}$ for the next boosting round (lines 12 and 13 of the Algorithm).
>
> **Specific section dedicated to the limitations of the methods proposed**
>
> In the submitted manuscript, the limitations of the methods proposed in terms of training complexity are described on lines 279-306 and lines 658-668. The new section dedicated limitations will state the additional computational time required by the column generation method in comparison with other existing approaches, and the related limitation in the usage of datasets with millions of samples.

---

> ### Comment · Reviewer_5GJn · 2025-08-01
> **rebuttal acknowledgement**
>
> Dear Authors,
>
> I appreciate the description of algorithm 1, and addressing the limitations section.
>
> The notation is challenging for me. The trouble is not about specific symbols, which are well-defined if I hunt for them (recent trends seem to push tables of symbols out of ML papers, but to me this would be helpful). When asking for an intuitive description of 5 and 7, the authors again give a statement in notation. I am looking for a natural language description here.
>
> I maintain that there tends to be no high-level description of the theorems outside of the symbols, e.g. like Algorithm 1 as given in the rebuttal. I'm happy just acknowledging that my assessment is largely high-level and low confidence. Whatever my understanding, I would expect authors to give a natural language description where it would benefit. So for now I'll maintain my weak reject score (with low confidence).

---

> > ### Author Response · Authors · 2025-08-01
> >
> > We thank the Reviewer for the prompt feedback received. We agree with the Reviewer about the relevance of intuitive explanations and tables with symbols. However, it is sometimes not easy to combine such explanations with multiple results under space constraints.
> >
> > We thought that the main problem was about the symbols used. In the following, we provide a high-level intuitive description of the uncertainty set in (5), the minimax rule (7), and Theorem 1.
> >
> > The uncertainty set in (5) is composed by all the probability distributions over instance-label pairs that are similar to the empirical distribution of training samples, as is commonly done in DRO methods. While other DRO methods define such similarity in terms of metrics such as Kullback-Leibler divergence or Wasserstein distance, the proposed methods define the similarity using the set of base-rules (e.g., the set of decision trees). Specifically, the proposed methods regard two distributions as similar if the expectations of all base-rules change only slightly when computed under either distribution. This notion of similarity is adequate because it leads to quite reduced uncertainty sets, since the set of base-rules is large. In addition, such definition allows us to obtain strong theoretical results due to the uniform convergence of expectations for base-rules.
> >
> >
> > Regarding Theorem 1 and the minimax rule in (7): To obtain classification rules that solve the minimax problem formulated in (4) is quite appropriate to provide robustness to noise, since these rules minimize worst-case error probabilities. However, obtaining such minimax classification rules addessing (4) in practice may seem to be quite difficult or imposible. Theorem 1 shows that indeed the minimax problem in (4) can be tractably addressed in practice because it is equivalent to the linear optimization problem in (6). In addition, such a result shows how the solution of the linear optimization problem provides a solution for the minimax problem in (4). In particular, the solution of the minimax problem shown in (7) is given by a combination of the base-rules with coefficients given by the solution of the linear optimization problem in (6).
> >
> > We hope these more natural explanations help the Reviewer to have a more clear idea of the results in the paper. We would be happy to provide additional explanations in natural language during the Authors/Reviewers discussion period.

---

> > > ### Comment · Reviewer_5GJn · 2025-08-01
> > >
> > > Dear authors,
> > >
> > > Thank you for your explanation. I'm happy to change my review to a weak accept. I wish you the best of luck in the paper's acceptance.

---

> > > > ### Author Response · Authors · 2025-08-04
> > > >
> > > > We would like to thank the Reviewer for his/her appreciation of the responses provided and for the support provided for the paper's acceptance

---

### Official Review · Reviewer_TwM6 · 2025-07-03

**Clarity:** 3
**Significance:** 3
**Originality:** 3
**Rating:** 5
**Confidence:** 3

**Summary:**

The paper presents a novel boosting framework designed to improve robustness against general types of label noise, called Robust Minimax Boosting (RMBoost). Existing boosting methods provide theoretical robustness guarantees for symmetric and uniform/non-uniform label noise, but they only offer bounds in expectation, not for finite samples. Moreover, the accuracy of these methods often drops significantly. RMBoost, instead, aims to be robust while also maintaining high accuracy. It directly minimizes the worst-case error probabilities over an uncertainty set defined by the training samples and base rules—that is, the base models of the boosting algorithm, where empirical averages can deviate from expected values within a threshold. The paper proposes an alternative formalization of the problem, whose solution is a classifier formed by a linear combination of the base rules. Moreover, the resulting classifier is proven to be optimal for minimizing the worst-case error, and the paper provides and proves a bound on its maximum error in the presence of label noise. The paper also proposes an efficient column generation algorithm for learning RMBoost rules. The experimental evaluation considers four state-of-the-art training techniques, four robust training techniques, and eight relatively small tabular datasets suited for training boosted models, evaluating both symmetric and uniform label noise as well as adversarial label noise. The experiments demonstrate RMBoost’s higher resilience to label noise and its comparable performance to standard boosting algorithms in the absence of noise.

**Questions:**

- Why was this baseline [A] not included in the experimental evaluation? Can you add it?
- Could you explain why larger and more realistic datasets such as Covtype, HIGGS, or SUSY were not included and if you plan to evaluate you approach on them?

**Update after rebuttal**
The authors have clarified the omission of baseline [A] in the experimental evaluation, explaining that they attempted to include this baseline but its performance did not reach the same level as the one of the other baselines in the paper. Regarding the second question, the authors provided results on a limited portion of the suggested datasets, that support the effectiveness of their approach. They have also clearly stated the limitation of their method in handling large tabular datasets. I find this more than acceptable, given that the limitation is explicitly acknowledged in the paper.

**Ethical Concerns:**

["NO or VERY MINOR ethics concerns only"]

**Final Justification:**

I recommend the acceptance of this work. This paper is original, since it addresses the problem of finding boosted models that are robust to general label noise and not just specific types, while characterizing their robustness as a function of the noise and ensuring they remain accurate. The theoretical contributions are well-grounded (all proofs are provided in the Appendix) and they demonstrate the existence of a boosted model that minimizes the worst-case error and for which it is possible to theoretically characterize both robustness to general noise and accuracy. The experimental evaluation is adequate; it considers several baselines and datasets, and shows that the proposed model is more robust and sufficiently accurate compared to models proposed previously in the literature. Both the experimental evaluation and the clarity of the proposal have been improved during the rebuttal, where the authors provided additional experimental results supporting the robustness of their models against noise.

**Limitations:**

The paper should include a limitations section clearly stating the proposal's limitations, such as the additional computational time required compared to other training solutions like AdaBoost and the absence of evaluation on large datasets.

**Update after rebuttal**
The authors will add this section in the final version of their work.

**Paper Formatting Concerns:**

No formatting concerns to signal.

**Quality:**

3

**Strengths And Weaknesses:**

Thanks to the authors for submitting this interesting paper. I appreciate the effort to provide theoretically well-grounded algorithms for training machine learning classifiers—particularly boosted models, that are both accurate and robust to noise (random or adversarial), as I believe such properties are highly desirable in practical applications. I think this is a good paper that proposes a solid theoretical contribution, although I believe the experimental evaluation could be improved a bit. I detail below the strengths and weaknesses of the paper and hope that the authors can address my questions during the rebuttal phase.


## Strengths

Regarding novelty, this paper is certainly original, as it concerns the problem of finding boosted models that are robust to general label noise (not just specific types of noise), characterizing their robustness as a function of the noise, and ensuring they remain accurate. The theoretical contributions are well-grounded (all proofs are provided in the Appendix) and appreciable, since they demonstrate the existence of a boosted model that minimizes the worst-case error and for which it is possible to theoretically characterize both robustness to general noise and accuracy. Finally, the experimental evaluation is adequate, although it could be improved; it considers several baselines and datasets and shows that the proposed model is more robust and sufficiently accurate compared with previous works.

## Weaknesses

First, the experimental evaluation considers both uniform-symmetric and adversarial label noise. However, in Section 2.2, the paper mentions [A], which provides robustness results going beyond symmetric and uniform cases, including cases with symmetric non-uniform noise. This baseline is not considered in the experimental evaluation, even though integrating it would be very interesting since RMBoost is designed to handle general label noise. I suggest that the authors include this baseline or specifically explain why it was not considered.

Second, I suggest considering more realistic and larger datasets in the experimental evaluation, such as Covtype, HIGGS, and SUSY from the UCI Machine Learning Repository, which have also been previously used to evaluate tree ensembles (e.g., in [B]). This addition would make the experimental evaluation more convincing. If it is not possible to include these datasets, I kindly ask the authors to explain why.

## Minor
- line 107, page 3 -> should it be $\phi$ and not $\phi'$?
- line 156, page 4 -> "margins' distribution" -> "distribution of the margins"
- line 210, page 5 -> "base-rules' values" -> "values of the base rules"

[A] Ghosh et. al., Making risk minimization tolerant to label noise, Neurocomputing, 2015.

[B] Chen et. al., Robustness Verification of Tree-based Models, in NeurIPS 2019.

---

> ### Author Rebuttal · Authors · 2025-07-30
>
> We thank the Reviewer for his/her appreciation of the theoretically well-grounded algorithms and the solid theoretical contribution. We also thank the Reviewer for the suggestions and comments provided that are addressed below. We will be happy to answer any further question the Reviewer could have during the authors/reviewers discussion period.
>
> **Experimental comparison with the methods analyzed in reference [14]**
>
> As described by the Reviewer, the results in reference [14] show robustness results that go beyond the common symmetric and uniform noise. Specifically, such work shows that certain potential functions (e.g., sigmoid) are also robust to symmetric and non-uniform label noise (in cases where the Bayes risk is zero). The submitted paper describes the results in [14] as one of the few works that go beyond the symmetric and uniform noise. However, the methods described in [14] are not directly comparable experimentally with the methods proposed: the results in [14] are not oriented to boosting methods and the potential functions analyzed are non-convex. Specifically, the classification rules considered in [14] are linear rules and quadratic-kernel rules, not ensembles of base-rules. In addition, the reference [14] utilizes dedicated optimization algorithms to deal with the non-convexity of the potential function, which have an unclear generalization in boosting settings. Nevertheless, over the past few days, we have used the XGBoost library to implement boosting methods with the potential functions proposed in [14]. This library allows the use of customized potential/loss functions. However, the performance obtained was significantly worse than that of all the other methods. We guess this is due to the non-convexity of the potential functions in [14].
>
> **Additional experimental results with larger datasets**
>
> The specific learning algorithm proposed in Section 5 of the paper to solve the minimax problem (4) is based on column generation approach for linear optimization, similarly as other methods such as LPBoost. A known limitation of approaches based on column generation is that their complexity increases with the number of samples faster than other boosting methods such as AdaBoost (see discussion on lines 279-306 and running times in Figure 5 comparing AdaBoost, LPBoost, and the proposed RMBoost). Notice also that approaches based on column generation have not been implemented in the reference [B]. For instance, the size of the optimization problems solved by the column generation methods increases linearly with the number of samples. Therefore, the usage of such approaches with large scale datasets with millions of samples require specialized optimization methods such as parallel solvers, dual decompositions or core-set selection, which go beyond the scope of the submitted paper.  In the final version of the paper, we will extend the description of this topic in the paragraph about computational complexity in relation with the reference, and state this limitation in the new section for limitations.
>
> Nevertheless, we agree with the Reviewer that it is of interest to include additional experiments with larger datasets than those used in the submitted manuscript (up to 1k samples). Hence, we have carried out new experimental results with the datasets suggested by the Reviewer, where the limitations of the column generation methods have been addressed by using a subset of the datasets composed by 5,000 samples. The following table shows the results obtained for different types of noise.
>
> |           | Dataset   | RobustB  | AdaB     | LPB      | LogitB   | GentleB  | BrownB   | GBDT     | XGB$-$Q    | RMB      | Minimax  |
> |-----------|-----------|----------|----------|----------|----------|----------|----------|----------|----------|----------|----------|
> |           | Susy      | 24±1.7 | 24±1.8 | 30±2.2 | 24±2.0 | 25±1.7 | 23±1.7 | 25±1.8 | 24±1.6 | 23±2.0 | 23±0.3 |
> | Noiseless | Higgs     | 34±1.8 | 33±2.0 | 38±3.1 | 33±1.9 | 34±2.1 | 34±1.9 | 37±2.0 | 37±2.4 | 35±2.1 | 33±0.3 |
> |           | Forestcov | 20±1.8 | 20±1.8 | 27±1.7 | 17±1.1 | 20±1.7 | 20±1.5 | 26±2.0 | 33±1.7 | 22±1.6 | 22±0.2 |
> | | | | | | | | | | | | |
> |           | Susy      | 26±2.2 | 24±1.8 | 35±3.4 | 27±1.9 | 28±1.8 | 24±1.8 | 26±1.8 | 24±1.8 | 23±1.9 | 28±0.4 |
> | $P_{\text{noise}} = 10%$     | Higgs     | 35±2.1 | 35±1.9 | 41±1.4 | 36±2.1 | 37±2.3 | 35±2.2 | 39±2.2 | 38±2.4 | 35±2.4 | 36±0.5 |
> |           | Forestcov | 22±1.7 | 22±1.7 | 34±1.9 | 22±1.2 | 25±2.0 | 23±1.6 | 27±2.1 | 33±2.2 | 23±1.7 | 28±0.4 |
> | | | | | | | | | | | | |
> |           | Susy      | 27±2.0 | 26±2.0 | 38±2.2 | 30±2.1 | 32±2.0 | 25±1.9 | 29±1.6 | 25±1.7 | 24±2.0 | 32±0.5 |
> |  $P_{\text{noise}} = 20%$       | Higgs     | 37±2.3 | 37±2.0 | 46±3.3 | 38±2.3 | 39±2.4 | 37±2.3 | 41±2.9 | 39±2.7 | 36±2.2 | 39±0.6 |
> |           | Forestcov | 24±1.8 | 24±1.8 | 38±2.3 | 25±1.5 | 29±1.9 | 25±1.9 | 32±2.1 | 34±2.7 | 23±2.0 | 32±0.4 |
> | | | | | | | | | | | | |
> |           | Susy      | 32±2.0 | 32±2.1 | 43±2.7 | 34±2.0 | 35±2.0 | 31±3.3 | 34±1.6 | 33±2.9 | 32±2.3 | 28±0.5 |
> | Adversarial  $P_{\text{noise}} = 10%$   | Higgs     | 39±2.0 | 39±2.0 | 48±2.8 | 41±2.0 | 42±2.3 | 38±3.4 | 44±1.2 | 38±1.7 | 38±2.4 | 40±0.5 |
> |           | Forestcov | 28±1.9 | 28±2.0 | 40±2.2 | 28±2.1 | 29±2.3 | 28±2.0 | 27±1.7 | 28±2.3 | 28±2.1 | 31±0.5 |
> | | | | | | | | | | | | |
> |           | Susy      | 40±2.0 | 40±1.9 | 49±2.8 | 43±1.9 | 44±2.2 | 40±1.5 | 41±2.3 | 39±2.0 | 38±2.5 | 33±0.5 |
> |  Adversarial $P_{\text{noise}} = 20%$       | Higgs     | 49±2.1 | 50±2.2 | 49±3.4 | 49±2.7 | 49±2.4 | 49±2.8 | 49±0.7 | 49±4.1 | 48±3.3 | 46±0.6 |
> |           | Forestcov | 39±2.5 | 39±2.4 | 47±3.8 | 38±2.2 | 40±2.2 | 39±2.1 | 38±1.4 | 36±2.7 | 36±2.6 | 36±0.7 |
>
>
> The additional results show similar behavior as that in the experimental results of the submitted manuscript. The proposed methods achieve adequate performance without noise together with improved robustness with noisy labels.
>
> [B] Chen et. al., Robustness Verification of Tree-based Models, in NeurIPS 2019.
>
> **Specific section dedicated to the limitations of the methods proposed**
>
> We thank the Reviewer for the suggestion provided about a section dedicated to the methods’ limitations in terms of training complexity. In the submitted manuscript, the limitations are described on lines 279-306 and lines 658-668. As suggested by the Reviewer, the new section for limitations will state the additional computational time required by the column generation method in comparison with other existing approaches, and the related limitation in the usage of datasets with millions of samples (the final version of the paper will include the new experimental results shown above).
>
> **Minor comments**
>
> We thank the Reviewer for pointing few typos and suggestions to improve the writing that will be addressed in the final version of the paper. Regarding the usage of $\phi$ or $\phi'$ in line 107, the correct usage is $\phi'(0)<0$ since the condition required for a potential function is that the derivative at zero is negative, that is, the potential function should be decreasing at zero since positive margins should be preferred to negative margins (see condition 2 in Definition 1 of reference [10]).

---

> > ### Comment · Reviewer_TwM6 · 2025-08-03
> > **Thank you to the authors for their response**
> >
> > Dear authors,
> > thank you for your response. I found your clarification regarding [14] very helpful, and I apologize for the misunderstanding on my part. The new experimental results support the explanation provided in the paper. As you mentioned, I hope you will clearly state the limitation related to the difficulties in handling large datasets in the appropriate section on limitations.
> >
> > I continue to recognize the quality of your work and I will fully support its acceptance.

---

> > > ### Author Response · Authors · 2025-08-04
> > >
> > > We would like to thank the Reviewer for his/her appreciation of the paper and the responses provided. In the final version of the paper we will clearly state the method's limitations in a dedicated section, as described in the responses

---

### Official Review · Reviewer_L2j8 · 2025-07-05

**Clarity:** 3
**Significance:** 3
**Originality:** 3
**Rating:** 4
**Confidence:** 4

**Summary:**

The paper presents a novel method called Robust Minimax Boosting (RMBoost) that aims to enhance the robustness of boosting algorithms against label noise. The authors introduce a minimax approach that directly minimizes worst-case error probabilities and provides finite-sample performance guarantees. The paper also includes efficient algorithms for RMBoost learning and demonstrates through experiments that RMBoost not only withstands label noise but also achieves strong classification accuracy in practice.

**Questions:**

see my comments above.

**Ethical Concerns:**

["NO or VERY MINOR ethics concerns only"]

**Final Justification:**

Thanks for the authors' efforts on providing more results in the rebuttal. I remain my support to this paper :)

**Quality:**

3

**Strengths And Weaknesses:**

**Strengths**
(1) The introduction of RMBoost, which focuses on minimizing worst-case error probabilities, is a novel advancement in the field of robust boosting methods.

(2) The paper provides comprehensive theoretical guarantees, including finite-sample performance and Bayes consistency, which are well-supported by solid proofs.

(3) Experiments are conducted to show that RMBoost outperforms existing methods in the presence of noisy labels and maintains high classification accuracy without noise.

(4) The proposed learning algorithms are efficient and scalable, which can be applied to handle large datasets.

**Weaknesses**
(1) While the experiments are thorough, additional testing on more diverse and real-world datasets (e.g., UCI ML datasets) could further validate the robustness and applicability of RMBoost.

(2) The paper assumes certain types of label noise, and the robustness to other, less common types of noise does not seem to be fully explored. Other papers such as [1][2] from the reference list below, also discussed methods for handling other types of label noise, e.g., one-sided label noise. It would be great if the authors can discuss their differences and perhaps include them in the experiments if applicable.

**Reference**:
[1] Liu, et al. "An analysis of boosted linear classifiers on noisy data with applications to multiple-instance learning." 2017 IEEE International Conference on Data Mining (ICDM). IEEE, 2017.
[2] Luan, et al. "Multi-Instance Learning with One Side Label Noise." ACM Transactions on Knowledge Discovery from Data 18.5 (2024): 1-24.

---

> ### Author Rebuttal · Authors · 2025-07-30
>
> We thank the Reviewer for his/her careful reading of the paper and for the positive feedback and suggestions received. We believe we completely address below the comments from the Reviewer, and will be happy to answer any further question the Reviewer could have during the authors/reviewers discussion period.
>
> **Additional experimental results with more diverse datasets**
>
> The experimental results in the paper show results using 8 publicly available real-world datasets commonly used to compare boosting methods. Five of them are available in UCI repository and the rest in Kaggle website. The instance dimensionality and label proportions in the datasets vary significantly among the datasets. Following the comment from the Reviewer, we have added additional results with other types of datasets that have a larger number of samples. The following table shows the additional results obtained using 5,000 samples from the datasets Susy, Higgs, and Forestcov that can be found in UCI repository.
>
> |           | Dataset   | RobustB  | AdaB     | LPB      | LogitB   | GentleB  | BrownB   | GBDT     | XGB$-$Q    | RMB      | Minimax  |
> |-----------|-----------|----------|----------|----------|----------|----------|----------|----------|----------|----------|----------|
> |           | Susy      | 24±1.7 | 24±1.8 | 30±2.2 | 24±2.0 | 25±1.7 | 23±1.7 | 25±1.8 | 24±1.6 | 23±2.0 | 23±0.3 |
> | Noiseless | Higgs     | 34±1.8 | 33±2.0 | 38±3.1 | 33±1.9 | 34±2.1 | 34±1.9 | 37±2.0 | 37±2.4 | 35±2.1 | 33±0.3 |
> |           | Forestcov | 20±1.8 | 20±1.8 | 27±1.7 | 17±1.1 | 20±1.7 | 20±1.5 | 26±2.0 | 33±1.7 | 22±1.6 | 22±0.2 |
> | | | | | | | | | | | | |
> |           | Susy      | 26±2.2 | 24±1.8 | 35±3.4 | 27±1.9 | 28±1.8 | 24±1.8 | 26±1.8 | 24±1.8 | 23±1.9 | 28±0.4 |
> | $P_{\text{noise}} = 10%$     | Higgs     | 35±2.1 | 35±1.9 | 41±1.4 | 36±2.1 | 37±2.3 | 35±2.2 | 39±2.2 | 38±2.4 | 35±2.4 | 36±0.5 |
> |           | Forestcov | 22±1.7 | 22±1.7 | 34±1.9 | 22±1.2 | 25±2.0 | 23±1.6 | 27±2.1 | 33±2.2 | 23±1.7 | 28±0.4 |
> | | | | | | | | | | | | |
> |           | Susy      | 27±2.0 | 26±2.0 | 38±2.2 | 30±2.1 | 32±2.0 | 25±1.9 | 29±1.6 | 25±1.7 | 24±2.0 | 32±0.5 |
> |  $P_{\text{noise}} = 20%$       | Higgs     | 37±2.3 | 37±2.0 | 46±3.3 | 38±2.3 | 39±2.4 | 37±2.3 | 41±2.9 | 39±2.7 | 36±2.2 | 39±0.6 |
> |           | Forestcov | 24±1.8 | 24±1.8 | 38±2.3 | 25±1.5 | 29±1.9 | 25±1.9 | 32±2.1 | 34±2.7 | 23±2.0 | 32±0.4 |
> | | | | | | | | | | | | |
> |           | Susy      | 32±2.0 | 32±2.1 | 43±2.7 | 34±2.0 | 35±2.0 | 31±3.3 | 34±1.6 | 33±2.9 | 32±2.3 | 28±0.5 |
> | Adversarial  $P_{\text{noise}} = 10%$   | Higgs     | 39±2.0 | 39±2.0 | 48±2.8 | 41±2.0 | 42±2.3 | 38±3.4 | 44±1.2 | 38±1.7 | 38±2.4 | 40±0.5 |
> |           | Forestcov | 28±1.9 | 28±2.0 | 40±2.2 | 28±2.1 | 29±2.3 | 28±2.0 | 27±1.7 | 28±2.3 | 28±2.1 | 31±0.5 |
> | | | | | | | | | | | | |
> |           | Susy      | 40±2.0 | 40±1.9 | 49±2.8 | 43±1.9 | 44±2.2 | 40±1.5 | 41±2.3 | 39±2.0 | 38±2.5 | 33±0.5 |
> |  Adversarial $P_{\text{noise}} = 20%$       | Higgs     | 49±2.1 | 50±2.2 | 49±3.4 | 49±2.7 | 49±2.4 | 49±2.8 | 49±0.7 | 49±4.1 | 48±3.3 | 46±0.6 |
> |           | Forestcov | 39±2.5 | 39±2.4 | 47±3.8 | 38±2.2 | 40±2.2 | 39±2.1 | 38±1.4 | 36±2.7 | 36±2.6 | 36±0.7 |
>
> The table above with additional results shows similar behavior as that in the experimental results of the submitted manuscript. The proposed methods achieve adequate performance without noise together with improved robustness with noisy labels.
>
> **Generality of the label noise considered in the paper and one-sided label noise**
>
> We would like to emphasize that the label noise considered in the paper is totally general (lines 78-79), and includes the one-sided label noise in [1] [2]. Specifically, any label noise is described by a function $\rho_y(x)$ that quantifies the probability with which the label of an instance $x$ is flipped from $y$ to $-y$. In the case of one-side label noise, the function $\rho_y(x)$ satisfies that $\rho_{+1}(x)=0$ and $0\leq\rho_{-1}(x)\leq1$ for all $x$, for cases where only the negative samples are affected by noise; respectively, $0\leq\rho_{+1}(x)\leq1$ and $\rho_{-1}(x)=0$ for all $x$, for cases where only the positive samples are affected by noise. The results in the paper consider arbitrary functions $\rho_y(x)$ so that they are valid for any type of label noise. We thank the Reviewer for pointing out the type of noise “one-sided label noise” that will be described in the final version of the paper updating the lines 78-86 with the corresponding references.
>
> We agree with the Reviewer that incorporating one-sided label noise into the experimental results can further validate the methods proposed. So that we have carried out other additional results in which we use one-sided noise, both uniform and adversarial (only one class affected by noise). The following table shows these additional results with one-sided label noise, in experimental results as those shown in Figure 2 of the submitted manuscript.
>
>
> |          |          | Method | 3%       | 6%       | 9%       | 12%       | 15%      | 18%       | 21%      | 24%      | 27%      | 30%      |
> |----------|----------|--------|----------|----------|----------|----------|----------|----------|----------|----------|----------|----------|
> | Uniform|    Diabetes       | AdaB   | 27±4.5 | 27±5.5 | 29±5.2 | 30±5.2 | 31±5.2 | 32±5.6 | 33±5.4 | 34±5.2 | 35±5.5 | 38±5.6 |
> | noise | | RMB    | 26±4.8 | 26±5.0 | 27±4.8 | 27±5.3 | 27±4.4 | 29±4.9 | 29±5.4 | 30±5.7 | 32±5.6 | 35±5.3 |
> | | | | | | | | | | | | |
> | Adversarial |    Diabetes       | AdaB   | 32±5.8 | 34±4.7 | 34±4.5 | 35±4.7 | 37±4.2 | 44±4.3 | 45±4.3 | 46±4.2 | 47±4.2 | 48±4.2 |
> | noise | | RMB    | 26±4.5 | 26±5.7 | 27±4.9 | 32±6.5 | 35±4.0 | 36±4.7 | 36±5.6 | 38±5.2 | 40±4.3 | 43±4.8 |
> | | | | | | | | | | | | |
> | Uniform|  Raisin          | AdaB   | 15±3.5 | 16±3.7 | 16±3.7 | 17±4.1 | 19±3.9 | 20±3.8 | 20±4.3 | 22±4.5 | 23±4.2 | 25±3.7 |
> | noise |          | RMB    | 15±3.4 | 15±3.6 | 15±4.0 | 15±3.9 | 15±3.6 | 16±2.4 | 17±4.1 | 17±4.0 | 18±4.3 | 19±4.5 |
> | | | | | | | | | | | | |
> | Adversarial |    Raisin       | AdaB   | 17±3.5 | 18±3.7 | 21± 3.9 | 22±3.7 | 24±3.7 | 24±3.5 | 26±3.9 | 31±4.1 | 31±3.9 | 31±3.7 |
> | noise | | RMB    | 14±3.9 | 17±2.7 | 17±2.6 | 18±3.9 | 19±3.4 | 19±2.7 | 20±5.0 | 23±4.7 | 24±3.7 | 27±4.8 |
>
> The results above shows that the proposed techniques are also robust to one-side label noise.

---

### Decision · Program_Chairs · 2025-09-17

**Decision:**

Accept (poster)

**Comment:**

This work develops a boosting method, directly minimizing worst-case error probabilities, that is tolerant to label noise. They provide an efficient implementation and exhibit experimentally that their algorithm performs well in practice. After some discussion, the reviewers agreed that this work is above the acceptance threshold, hence I recommend acceptance.